# EFFICIENT SEGMENTATION USING ATTENTION-FUSION MODULES

## ABSTRACT

Fusing global and local semantic information in segmentation networks remains challenging due to computational costs and the need for effective long-range recognition. Based on the recent success of transformers and attention mechanisms, this research applies attention-based methods of attention-boosting modules and attention-fusion networks in enhancing the performance of state-of-the-art segmentation networks, such as InternImage and SERNet-Former, addressing these challenges. Integrating attention-boosting modules into residual networks generates baseline architectures like Efficient-ResNet, enabling them to extract global context feature maps in the encoder while minimizing computational costs. Attention-based algorithms can also be applied to networks utilizing vision transformers and convolutional layers, such as InternImage, to improve the existing results of state-of-the-art networks. In this research, SERNet-Former is deployed on the challenging benchmarking datasets such as ADE20K, BDD100K, CamVid, and Cityscapes by depending on the attention-based methods with new implementations of the network, SERNet-Former_v2. Our methods have also been implemented for InternImage-XL and improved the test performance of the network on the Cityscapes dataset (85.1 % mean IoU). Respectively, the results of the selected networks developed by our methods on the challenging benchmarking datasets are found worth considering: 85.1 % mean IoU on the Cityscapes test dataset, 59.35 % mean IoU on ADE20K validation dataset, 67.42 % mean IoU on BDD100K validation dataset, and 84.62 % mean IoU on the CamVid dataset.

## 1 INTRODUCTION

Segmentation is a widely applied computational task for scene understanding in the field of computer vision. Each pixel or mask in an image is represented through the labeled semantic classes in semantic segmentation. The labeled classes represent the ground truth of an image input predicted by state-of-the-art networks and methods. Semantic segmentation has numerous applications, including autonomous driving, robotics for indoor and outdoor scene recognition, medical imaging, virtual and augmented reality, real-time surveillance, and photography (Minaee et al., 2022; Borse et al., 2021; Huang et al., 2023; Erişen, 2024). Segmentation networks are developed through Fully Connected Networks (FCNs), Convolutional Neural Networks (CNNs), Vision Transformers (ViT), as well as Swin Transformers with shifted windows attention mechanisms and mostly based on the encoder-decoder architectures (Borse et al., 2021; Huang et al., 2023; Wang et al., 2023; Du et al., 2022; Xie et al., 2021; Liu et al., 2021; Chen et al., 2023; Erişen, 2024). Encoder-decoder architectures have shown remarkable progress, and recently, ViT-based networks with attention mechanisms achieved state-of-the-art performances on semantic segmentation datasets (Wang et al., 2023; Erişen, 2024).

Despite advancements, achieving efficient multi-scale feature fusion and overcoming computational bottlenecks remain significant limiting factors in segmentation tasks. The challenge of recognizing objects in segmentation is two-fold: The labeled object may either lose the spatial information or feature-rich properties while processing the image throughout the network (Erişen, 2024). Recent state-of-the-art networks (Borse et al., 2021; Wang et al., 2023; Chen et al., 2018; Erişen, 2024) seek to resolve the discrepancies between the semantic information extracted from the global and local contexts of a network.

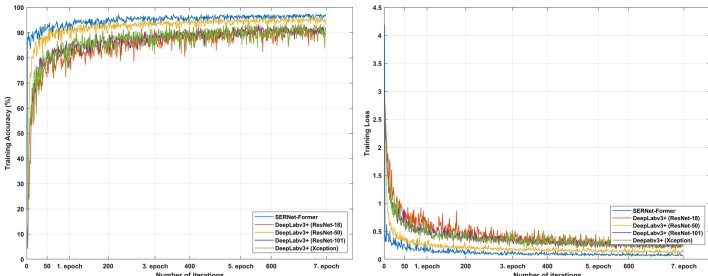

Figure 1: Training progress of SERNet-Former compared to the selected baselines on the CamVid dataset.

It is another fact that increasing the number of convolution layers only sometimes returns efficient results when compared to its computational cost (Wu et al., 2019). Recent research has moved away from solely expanding convolutional neural networks and instead explores their integration with attention mechanisms improving performance (Erişen, 2024). Hence, this study re-evaluates encoder-decoder architectures combined with attention-based fusion modules. It is aimed to investigate the spatial information and feature maps thoroughly within convolutional neural networks with attention mechanisms to enable efficient and accurate semantic segmentation. For instance, SERNet-Former (Erişen, 2024) explores fused attention-gates and skip connections to optimize spatial and channel-based feature extraction within both encoder and decoder stages. InternImage (Wang et al., 2023), on the other hand, deploys deformable convolution layers that work in harmony with the deformable attention networks and transformer heads improving the network's performance.

In this research, the residual networks are analyzed first as lightweight baselines for efficient training by considering the training accuracy and loss as the initial performance parameters in the network design with attention-based methods (Fig. 1) (He et al., 2016; Erişen, 2024). Different activation functions and attention gates are examined first with the residual convolution layers in the encoder and decoder parts. Attention gates, strategically integrated into SERNet-Former, enhance feature extraction, and the designed skip connections ensure the seamless fusion of multi-scale features (Erişen, 2024). Experimental results demonstrate the effectiveness of SERNet-Former in leveraging the fused spatial and semantic information (Fig. 1) (Chen et al., 2018; 2017; Erişen, 2024).

Respectively, the encoder of SERNet-Former is found as the most compatible baseline to be developed and improved with the attention-boosting gates (AbGs) and attention-boosting modules (AbMs) resulting in an efficient residual network, Efficient-ResNet, with the increased training performance and prediction efficiency (Erişen, 2024). Building on similar objectives of combining convolutional layers with attention mechanisms, this research also explores InternImage architectures. AbGs are incorporated as activation functions in InternImage variants, such as InternImage-XL, leading to significant performance improvements. However, these enhancements come at the cost of increased computational demands. Attention-fusion networks (AfNs) are incorporated into the decoder of SERNet-Former (Erişen, 2024) to enhance its functionality by effectively storing, fusing, and processing rich semantic information from the encoder during the up-sampling phase. These networks are purposefully designed to blend global spatial context with feature-based rich semantic information (Erişen, 2024). Consequently, our methods, AbG, AbM, and AfN, exhibit significant improvements with the residual networks employed in SERNet-Former and SERNet-Former_v2, which is recompiled in PyTorch, along with notable advancements in the InternImage-XL network.

To summarize, attention-boosting gates, modules, and attention-fusion networks are utilized within encoder-decoder architectures, such as SERNet-Former and InternImage (Erişen, 2024; Wang et al., 2023). Attention-Boosting Gates (AbGs), Attention-Boosting Modules (AbMs), and Attention-Fusion Networks (AfNs) integrate rich information from various contexts by combining essential semantic data to achieve maximum efficiency. The attention-boosting modules improve the performance of the networks by leveraging the feature-rich information of the global context. Attention-fusion networks help overcome long-range dependencies in predicting and recognizing smaller ob-

jects by retaining their features. Our methods deployed on the selected networks return impressive results on the ADE20K, BDD100K, CamVid and Cityscapes datasets (Brostow et al., 2008; 2019; Cordts et al., 2016; Zhou et al., 2017; Yu et al., 2020). Our contributions are briefly highlighted:

- The attention-boosting gates (AbGs) and modules (AbMs) are applied to segmentation networks with residual layers, such as SERNet-Former, as well as networks with deformable convolutional layers and attention heads, such as InternImage for the increased performance

- The capacity of the decoder of the SERNet-Former is improved via attention-fusion networks (AfNs), increasing the efficiency of leveraging the pixel-wise and feature-rich semantic information from the local and global contexts

- Skip connections provided efficient connections in fusing and concatenating the multi-scale information from the global and local contexts

- SERNet-Former is reimplemented in PyTorch, SERNet-Former_v2, based on the applied methods

- The networks that apply our methods achieved state-of-the-art performances on the CamVid dataset, Cityscapes, and BDD100K validation datasets.

## 2   RELATED WORKS

The multi-scale problem in computer vision refers to the challenge of integrating spatial and channel-based semantic information of an object in segmentation networks from both global and local contexts (Erişen, 2024). DeepLabv3+ (Chen et al., 2018) is the widely known segmentation architecture developed to fuse the feature-based rich semantic data with the spatial information, similar to other encoder-decoder networks like U-Net and Segnet (Badrinarayanan et al., 2017; Foroughi et al., 2021). The recent success of transformers integrated with CNNs (Wang et al., 2023) also revealed that additional methods, such as transformer heads and attention mechanisms (Yan et al., 2024) improve the efficiency of networks concerning multi-scale representations.

Respectively, Li et al. (2022a) introduced global enhancement and local refinement methods integrated through a Context Fusion Block against the challenge of fusing the global and local semantic information and loss of features during down-sampling and up-sampling (Li et al., 2020). The Guided Attention Inference Network (GAIN) developed by Li et al. (2020) employs fully convolutional networks and CNN-based semantic segmentation architectures, integrating widely utilized baselines such as ResNet-101 (He et al., 2016; Xie et al., 2017).

To enhance the synthesis between global and local semantic information alongside spatial and channel-wise features, the Squeeze-and-Excitation block (SENet), introduced by Hu et al. (2020), has been implemented. In a similar vein, self-attention mechanisms have been designed to extract comprehensive feature information from objects at each position by aggregating features from all locations within a single sample (Guo et al., 2023). Consequently, Guo et al. (2023) proposed an external attention module that employs memory units, thus replacing the conventional self-attention mechanisms within semantic segmentation networks. The influences of SENet and self-attention mechanisms have also inspired the development of SAB Net, which presents an end-to-end semantic attention-boosting framework (Ding et al., 2022). This methodology proposes a non-local semantic attention framework that regularizes the discrepancies between non-local and local information by applying category-wise learning weights (Ding et al., 2022). CTNet provides an alternative strategy utilizing a Channel Contextual Module to explore multi-scale local channel contexts and a Spatial Contextual Module to examine global spatial dependencies in a combined configuration (Li et al., 2022b).

Additionally, CoTNet integrates transformer architectures with self-attention mechanisms by incorporating a Contextual Transformer Block, which facilitates the transformation of each discrete convolutional operator (such as 3 by 3 convolutions) into two consecutive 1 by 1 convolutions (Li et al., 2023). Furthermore, Ye et al. (2022) presented cross-modal self-attention, which was designed to integrate image and language expressions as inputs.

Attention-based feature fusion has emerged as an effective approach addressing the multi-scale challenges inherent in semantic segmentation (Yang & Gu, 2023). This technique employs across-feature

maps to tackle the difficulties associated with small objects, which are often difficult to accurately identify due to their semantic characteristics and precise spatial information (Sang et al., 2023). Furthermore, Choe et al. (2021) proposed an attention-based dropout layer designed to obscure the most discriminative features of the model by utilizing a drop mask and an importance map, all while preserving classification accuracy.

As an alternative to attention-fusion networks, Liu et al. (2022) introduced the covariance attention method, which employs the covariance matrix to delineate the dependencies between local and global semantic features. In a related study, Yang et al. (2021a) developed a variational structured attention mechanism to integrate channel-based and spatial features. Their methodology produces tensor products derived from spatial and channel-wise attention modules, facilitating the evaluation of the probabilities of latent variables that connect these two attention mechanisms. Additionally, Hao et al. (2022) proposed a technique known as spatial-detail-guided context propagation, which seeks to reconstruct lost information in low-resolution global contexts by leveraging the spatial details from shallower layers in real-time. The integration of spatial information with channel-based rich semantic features has also been examined through the lens of RGB-D networks and 3D point clouds (Cao et al., 2021; Chen et al., 2020; Jian et al., 2021; Yang et al., 2021b). Moreover, Huang et al. (2023) introduced CCNet, a crisscross attention network that presents a novel alternative to conventional attention mechanisms. Respectively, this research evaluates the potential of convolutional networks with attention-based mechanisms to be applied for segmentation tasks.

## 3 METHOD

This research aims to assess the attention-based mechanisms of attention-boosting gates (AbGs), modules (AbMs), and attention-fusion networks (AfNs) in improving efficient segmentation networks. Attention-boosting gates (AbGs) and attention-boosting modules (AbMs) are designed to excite and fuse the feature-rich spatial information into AfNs as well as the existing networks, such as ResNet, for fast and accurate training without losing the progress (Erişen, 2024). Accordingly, the state-of-the-art networks, SERNet-Former and InternImage, using encoder-decoder architectures with the same motivation using convolution layers and attention-based mechanisms (Wang et al., 2023; Erişen, 2024) are deployed for the experiments and analyses of these attention-based methods.

Residual networks pre-trained on the ImageNet dataset serve as robust baselines for segmentation tasks due to their efficiency in learning key features rapidly, despite capacity constraints for novel features (Erişen, 2024). Respectively, SERNet-Former (Erişen, 2024) is found as the most efficient network to deploy AbGs, AbMs, as AfNs in this research (Fig. 2). AbGs utilize attention-based algorithms to enhance the extraction of equivariant, pixel-wise, feature-rich semantic information from selected baselines, maintaining equivalent input and output sizes (Erişen, 2024). Thus, the feature-rich semantic information is integrated with the spatial context of the encoder through AbMs. In the modification of InternImage architectures, however, the activation function is replaced with the algorithms deploying the attention-boosting gates together with the ReLU function in this research.

To further improve feature learning and semantic representation, we explore attention-based fusion modules integrated into convolutional neural networks. These modules address limitations in smaller networks by enhancing spatial and channel-based information processing through attention mechanisms. Therefore, AbMs and AfNs, deployed in SERNet-Former (Erişen, 2024), amplify signal efficiency and fuse spatial and channel-based semantics during decoding, effectively addressing capacity issues in smaller residual networks. In InternImage, AbGs and AbMs are introduced as a layer of adaptive and attention-centric design replacing the traditional activation function. Dilation-based network (DbN) was also applied to SERNet-Former architectures (Chen et al., 2018; Erişen, 2024) in facilitating the efficient transmission of local semantic patterns between encoder and decoder components. DbN supports the seamless transfer of spatially rich features while preserving computational efficiency. SERNet-Former_v2 is developed in PyTorch, based on the methods applied in SERNet-Former together with attention mechanisms. In brief, the methods applied in improving the efficiency of residual networks and InternImage architectures in this research are highlighted as follows:

- AbgS and AbMs are used as the attention-based activation functions in segmentation networks, such as SERNet-Former and InternImage-XL

Figure 2: The schematic illustration of applied methods on SERNet-Former. (a) Attention-boosting Gate (AbG) and Attention-boosting Module (AbM) are fused into the encoder part. (b) Attention-fusion Network (AfN), introduced into the decoder

- DbN bridges the encoder and decoder in SERNet-Former, which is improved by AfNs with the help of skip connections
- AbGs, AbMs, AfNs, and DbNs are applied to SERNet-Former_v2 in PyTorch with attention-based algorithms and transformers

## 3.1 ATTENTION-BOOSTING GATES

Leveraging the feature-based semantic information from the global context of segmentation networks confronts the increasing computational cost of attention heads. In this research, attention-boosting gates and modules are designed to excite and fuse feature-rich semantic information by applying attention-based algorithms to deal with this problem. In that regard, the Sigmoid function, widely applied in attention networks, is deployed as the activation function for attention-boosting gates, which excite pixel-wise rich semantic information that may not be activated through ReLU layers. Thus, it is aimed at extracting the feature-rich scalar values of each pixel from the data processed through the convolutional layers of residual networks instead of generating larger global attention maps. AbGs are initially designed alongside attention mechanisms and then modified into the weightless mathematical operators to be adapted to networks deploying attention transformers. Hence, AbGs decrease the computational cost significantly compared to the conventional transformer architectures without compromising the possibility of acquiring and processing equivariant, pixel-wise, and feature-based rich semantic information in proposing efficient attention mechanisms for residual networks. The use of the Sigmoid function in Equation (1), and the activation operation of the gate, AbG, can be iterated in Equation (2) as follows:

$$\sigma(AbG_i) = \frac{1}{1 + e^{-(BN(conv_n(i)))}}, \tag{1}$$

$$AbG_n = \sigma(AbG_i) \times (BN(conv_n(i))), \tag{2}$$

where $(BN(conv_n(i)))$ denotes the output of the last convolution and the following batch normalization layers processing the input $i$ at the $n$th convolution block. Thus, AbG derives the scalar values from the backbone architecture without resizing the product of inputs and the activated feature-rich scalar values. Hence, the height and width of the output of AbG are kept equivalent to the sizes of the transformed output of the convolution layers.

## 3.2 ATTENTION-BOOSTING MODULES

The generated feature-rich maps in AbGs are forwarded next as the product of weights and activation function in the attention-boosting modules, AbMs. Equation (3) illustrates the principles applied in processing the fused semantic information throughout AbM:

$$AbM_n = softmax(AbG_n + b(BN(conv_n(i)))), \tag{3}$$

where $b(BN(conv_n(i)))$ denotes the biases that are processed and forwarded in AbGs acquired from the input. AbMs fuse the forwarded feature-based semantic information with the spatial context of the networks, as illustrated in Equation (4) and Fig. 2 (a), after transforming the inputs.

$$AbM_{n,output} = AbM_n \bigoplus ReLU(BN(conv_n(i))), \tag{4}$$

$\bigoplus$ stands for the fusion function by concatenation. AbMs are added to the baseline at the end of each $n$th convolution block in SERNet-Former. Thus, a novel and efficient residual network architecture, Efficient-ResNet, has been developed (Fig. 2) (Erişen, 2024).

Skip connections are designed for efficient multi-scale feature fusion in SERNet-Former's encoder and decoder, preventing gradient vanishing of the Sigmoid function connected to the residual layers via AbMs. AbMs work as attention mechanisms and mathematical operators for exciting and fusing feature-rich semantic information. AbMs are also introduced into SERNet-Former_v2, and InternImage-XL by changing its activation function and replacing the ReLU function with AbGs yet reinforced again with ReLU to prevent the gradient vanishing while leveraging the feature-based semantic information.

### 3.3 Attention-fusion networks

The attention-based algorithms of AbG and AbM are also designed to be applied in different attention mechanisms and networks. Thus, they are deployed in the attention-fusion networks in the decoder part of SERNet-Former (Erişen, 2024). The initial layers of CNNs contain rich global semantic information, featuring sharp edges and distinct shapes of objects (Erişen, 2024). Thus, efficient up-sampling in the decoder relies on effectively transferring and reconstructing the matrix weights and spatially rich features from the encoder processed through attention-based transformers. This ensures precise one-to-one image processing during semantic segmentation tasks (Erişen, 2024). Thus, attention-based semantic information from the global and local contexts in the decoder part is fused by attention-fusion networks leveraging the pixel-wise scalar values activated through the Sigmoid function.

To address the limited feature-learning capacity of smaller residual networks, AfNs are designed with additional convolution layers (Fig. 2(b)). These layers enhance the decoder's ability to store and process semantic information. AfNs combine spatial and channel-based contextual features derived from deconvolution layers with varying strides. The resulting outputs are fused through a depth concatenation layer for further refinement (Fig. 2). Skip connections are also employed to streamline spatial information transfer from the encoder and integrate these features with channel-based information during up-sampling operations.

## 4 Experiments and results

This section begins by presenting the selected experimental datasets along with the corresponding implementation details. Subsequently, the results for each open-source dataset are thoroughly analyzed to facilitate a comparison with state-of-the-art models in the field. Furthermore, the impact of the methods employed in this analysis is discussed. Ablation studies are conducted as part of a comprehensive analysis to dissect and understand each method's contribution meticulously.

### 4.1 Datasets

**The Cambridge-driving Labelled Video Database (CamVid)** (Brostow et al., 2008; 2019) is a pioneering resource for scene understanding, specifically designed for semantic segmentation tasks. It comprises 701 images (720 by 960 pixels) captured from five video sequences using fixed CCTV-style cameras mounted on a vehicle. The dataset initially featured 32 annotated classes, later consolidated into 11 classes for practicality. The standard dataset split includes 367 training, 101 validation, and 233 test images, as commonly used in literature.

Table 1: Class accuracies (mIoU) of state-of-the-art methods on the CamVid dataset

| Reference | Method | Building | Tree | Sky | Car | Sign | Road | Pedestrian | Fence | Pole | Sidewalk | Bicycle | mIoU |
|---|---|---|---|---|---|---|---|---|---|---|---|---|---|
| CVPR 2018 | VideoGCRF | 86.1 | 78.3 | 91.2 | 92.2 | 63.7 | 96.4 | 67.3 | 63.0 | 34.4 | 87.8 | 66.4 | 75.2 |
| CVPR 2019 | Zhu et al. (2019) | 91.2 | 83.4 | 93.1 | 93.9 | 71.5 | 97.7 | 79.2 | 76.8 | 54.7 | 91.3 | 79.7 | 82.9 |
| CVMI 2024 | **SERNet-Former** | **93.0** | **88.8** | **95.1** | 91.9 | **73.9** | 97.7 | 76.4 | **83.4** | **57.3** | 90.3 | **83.1** | **84.6** |

Table 2: Per-class accuracies (mIoU) based on Cityscapes test dataset

| Groups / Labels | flat | | construction | | | object | | | nature | | sky | person | | vehicle | | | | | | |
|---|---|---|---|---|---|---|---|---|---|---|---|---|---|---|---|---|---|---|---|---|
| Method | road | sidewalk | building | wall | fence | pole | traffic light | traffic sign | vegetation | terrain | sky | person | rider | car | truck | bus | train | motorcycle | bicycle | mIoU |
| DeepLabv3 | 98.6 | 86.2 | 93.5 | 55.2 | 63.2 | 70.0 | 77.1 | 81.3 | 93.8 | 72.3 | 95.9 | 87.6 | 73.4 | 96.3 | 75.1 | 90.4 | 85.1 | 72.1 | 78.3 | 81.3 |
| DeepLabv3+ | 98.7 | 87.0 | 93.9 | 59.5 | 63.7 | 71.4 | 78.2 | 82.2 | 94.0 | 73.0 | 95.8 | 88.0 | 73.0 | 96.4 | 78.0 | 90.9 | 83.9 | 73.8 | 78.9 | 82.1 |
| DPC | 98.7 | 87.1 | 93.8 | 57.7 | 63.5 | 71.0 | 78.0 | 82.1 | 94.0 | 73.3 | 95.4 | 88.2 | 74.5 | 96.5 | 81.2 | 93.3 | 89.0 | 74.1 | 79.0 | 82.7 |
| Panoptic DeepLab | 98.8 | 88.1 | 94.5 | 68.1 | 68.1 | 74.5 | 80.5 | 83.5 | 94.2 | 74.4 | 96.1 | 89.2 | 77.1 | 96.5 | 78.9 | 91.8 | 89.1 | 76.4 | 79.3 | 84.2 |
| EfficientPS | 98.8 | 88.2 | 94.3 | 67.6 | 67.7 | 73.4 | 80.2 | 83.3 | 94.3 | 74.4 | 96.0 | 88.7 | 75.3 | 96.6 | 83.5 | 94.0 | 91.1 | 73.5 | 79.7 | 84.2 |
| iFLYTEK-CV | 98.8 | 88.4 | 94.4 | 68.9 | 68.9 | 73.0 | 79.3 | 83.3 | 94.3 | 74.3 | 96.0 | 88.0 | 76.3 | 96.6 | 84.0 | 91.7 | 92.6 | 74.7 | 79.3 | 84.5 |
| HRNet+OCR+SegFix | 98.9 | 88.3 | 94.4 | 68.0 | 67.8 | 73.6 | 80.6 | 83.9 | 94.4 | 74.5 | 96.1 | 89.2 | 75.9 | 96.8 | 83.6 | 94.2 | 91.3 | 74.0 | 80.0 | 84.5 |
| ViT-Adapter-L | 98.9 | 88.5 | 94.5 | 66.7 | 70.2 | 74.5 | 80.2 | 83.6 | 94.4 | 73.7 | 96.2 | 89.7 | 79.0 | 96.7 | 85.5 | 94.4 | 90.5 | 79.7 | 81.8 | 85.2 |
| InternImage-H | 98.9 | 88.8 | 94.9 | 72.5 | 71.2 | 75.4 | 80.9 | 84.7 | 94.5 | 75.5 | 96.3 | 90.1 | 79.9 | 96.8 | 85.3 | 95.5 | 92.6 | 80.0 | 82.2 | 86.1 |
| SERNet-Former* | 96.8 | 76.3 | 90.0 | 57.2 | 54.6 | 52.9 | 60.5 | 66.0 | 90.9 | 64.6 | 93.9 | 79.0 | 61.6 | 93.5 | 69.7 | 85.3 | 74.7 | 59.7 | 65.6 | 73.3 |
| SERNet-Former** | 97.8 | 81.3 | 91.0 | 60.6 | 57.3 | 57.4 | 64.1 | 70.7 | 91.6 | 66.9 | 94.7 | 80.9 | 65.2 | 94.3 | 80.4 | 90.6 | 86.8 | 63.7 | 68.8 | 77.0 |
| **SERNet-Former†** | 98.2 | 90.2 | 94.0 | 67.6 | 68.2 | 73.6 | 78.2 | 82.1 | 94.6 | 75.9 | 96.9 | 90.0 | 77.7 | 96.9 | 86.1 | 93.9 | 91.7 | 70.0 | 82.9 | 84.8 |
| **SERNet-Former_v2††** (ours) | 98.8 | 88.3 | 94.6 | 72.6 | 69.5 | 73.3 | 78.7 | 83.2 | 94.2 | 74.7 | 96.2 | 88.9 | 76.6 | 96.5 | 84.3 | 95.3 | 92.7 | 77.0 | 79.8 | 85.0 |
| InternImage-XL | 98.9 | 88.7 | 94.7 | 72.1 | 70.3 | 73.4 | 79.1 | 83.5 | 94.3 | 74.5 | 96.1 | 88.9 | 76.1 | 96.7 | 84.2 | 94.7 | 91.1 | 75.0 | 79.8 | 84.8 |
| **InternImage-XL†††** (ours) | 98.9 | 88.7 | 94.7 | 72.8 | 70.2 | 73.4 | 79.1 | 83.5 | 94.3 | 74.5 | 96.2 | 88.9 | 76.2 | 96.7 | 85.0 | 95.2 | 92.4 | 76.2 | 79.9 | 85.1 |

\*: ResNet-50 baseline without AbM, DbN, AfN. \*\*: Efficient-ResNet, based on ResNet-101 without DbN, AfN. †: SERNet-Former with AbM, DbN, AfN.

††: SERNet-Former_v2, developed in PyTorch. †††: InternImage-XL with AbGs and AfNs

**Cityscapes** (Cordts et al., 2016) is a highly challenging dataset for urban scene segmentation, offering high-quality pixel annotations for 5000 images and coarse annotations for an additional 20000 images. Captured across 50 European cities in various seasons under fair weather conditions, these stereo images (1024 by 2048 pixels) provide diverse urban environments (Cordts et al., 2016). The dataset includes fine-grained annotations for 30 classes grouped into eight categories, though most research focuses on 20 classes, with 19 semantic labels and one void class for ambiguous regions. The 5000 fine annotations are divided into 2975 training, 500 validation, and 1525 test images.

**ADE20K** (Zhou et al., 2017) includes samples for scene parsing, instance segmentation, and panoptic segmentation with pixel-level annotations and masked object instances. The scene parsing examples around 25000 images include 22210 for training, 2000 for validation, and 3352 images for testing. The images from indoor and outdoor scenes, including urban, rural, residential, and natural environments, are annotated with 150 semantic categories.

**BDD100K** (Yu et al., 2020) is created with the purpose of developing autonomous driving research through the tasks of semantic segmentation, instance segmentation, panoptic segmentation, and object detection. It focuses on real-world driving scenarios, offering a comprehensive view of urban environments with diverse weather, lighting, and scene types. The dataset includes 100000 images with annotations, split into 70000 for training, 10000 for validation, and 20000 for testing annotated through 40 classes, even though the common works utilized the same 19 semantic classes used in the evaluation of the Cityscapes dataset.

## 4.2 IMPLEMENTATION DETAILS AND EXPERIMENT RESULTS

SERNet-Former was trained on the 11-class CamVid dataset for 80 epochs with the original resolution of images (720 by 960 pixels). In the dataset split, standard practices are followed with a mini-batch size of 3 and an initial learning rate of 0.001. The experiments were conducted using

Table 3: Results of state-of-the-art methods on Cityscapes datasets

| Reference | Method | Baseline architecture | *test mIoU* | *val mIoU* |
|---|---|---|---|---|
| PAMI 2023 | CCNet | ResNet-101 | 81.9 | 80.2 |
| ICCV 2019 | Gated-SCNN | WideResNet | 82.8 | 80.8 |
| CVPRW 2022 | ResNeSt | ResNeSt | 83.3 | 82.7 |
| ECCV 2018 | DeepLabv3 | ResNet-101 | 81.3 | 78.5 |
| ECCV 2018 | DeepLabv3+ | Dilated-Xception-71 | 82.1 | 79.6 |
| CVPR 2019 | Auto-DeepLab | Auto-DeepLab-L | 82.1 | 80.33 |
| NeurIPS 2018 | DPC | Xception | 82.7 | 80.85 |
| ECCV 2022 | kMaX-DeepLab | ConvNeXt-L | 83.2 | 83.5 |
| CVPR 2020 | Panoptic DeepLab | SWideRNet | 84.2 | 83.1 |
| IJCV 2020 | AdapNet++ | ResNet-50 | 81.3 | 81.2 |
| IJCV 2020 | SSMA | ResNet-50 | 82.3 | 82.2 |
| IJCV 2021 | EfficientPS | EfficientNet-B5 | 84.2 | 82.1 |
| ECCV 2020 | HRNetV2+OCR | HRNetV2-W48 | 83.7 | 86.3 |
| ECCV 2020 | HRNetV2+OCR+SegFix | HRNetV2-W48 | 84.5 | 86.95 |
| arXiv 2021 | HS3-Fuse | HRNet48-OCR-HMS | 85.7 | - |
| NeurIPS 2021 | SegFormer | MiT-B5(IM-1K, MV) | 83.7 | 84.0 |
| ICLR 2024 | Lawin+ | Swin-L (IM-22K) | 84.4 | - |
| CVPR 2024 | DepthAnything | ViT-L | 84.8 | 86.2 |
| CVMI 2024 | **SERNet-Former** | **Efficient-ResNet** | 84.8 | 87.35 |
| ICLR 2023 | ViT-Adapter-L | ViT-Adapter-L | 85.2 | 85.8 |
| CVPR 2023 | InternImage-H | Mask2Former | 86.1 | 87.0 |
| | **SERNet-Former_v2** (ours) | **Efficient-ResNet_v2** (ours) | 85.02 | 86.5 |
| CVPR 2023 | InternImage-XL | UperNet | 84.85 | 86.2 |
| | **InternImage-XL** (ours) | UperNet | 85.1 | 86.5 |

Table 4: Performance results of models developed and tested in PyTorch

| Dataset | Model | mIoU | inference time (s/task) | parameters (M) |
|---|---|---|---|---|
| ADE20K (2K validation) | | | | |
| | SERNet-Former_v2 (ours) | 59.35 | 0.75 | 245 |
| BDD100K (10K validation) | | | | |
| | SERNet-Former_v2 (ours) | 67.42 | 0.75 | 245 |
| Cityscapes (test) | | | | |
| | SERNet-Former_v2 (ours) | 85.02 | 0.75 | 245 |
| | InternImage-XL (ours) | 85.10 | 0.76 | 368 |

MATLAB® (Erişen, 2024). Comparative results, including per-class mIoU and performance metrics of state-of-the-art models, are presented in Tables 1 and 6 (Erişen, 2024).

For the Cityscapes dataset, training configurations included a mini-batch size of 1 for single-scale resolution and 4 for images resized to 512 by 1024 pixels. An initial learning rate of 0.0005 was applied. Efficient self-training methods were implemented to reduce training time, initially using 715 selected samples for a 20-class setup, later expanded to 19 classes with all available training samples. The model pre-trained on CamVid was further fine-tuned on Cityscapes for 80 epochs in MATLAB®. Performance metrics, including per-class mIoU, are reported in Tables 2 and 3, alongside state-of-the-art benchmarks (Erişen, 2024).

The class weights for each dataset were incorporated into the loss function, with stochastic gradient descent (SGD) as the optimizer (momentum = 0.9). L2 regularization was applied at varying levels to minimize loss and improve efficiency. The experiments in MATLAB are run using Intel® Core™ i5-6200 CPU at 2.30–2.40 GHz with 16 GB RAM and NVIDIA® GeForce graphics card with 10 GB GPU memory.

Additional implementations of SERNet-Former on the Cityscapes as well as ADE20K (Zhou et al., 2017) and BDD100K (Yu et al., 2020) datasets are also achieved in PyTorch, denoted as SERNet-Former_v2, with different hardware resources for the best practices of efficient and practical training. Similarly, InternImage-XL is also tested and re-trained based on our applied methods on the challenging Cityscapes dataset. The results of InternImage-XL and SERNet-Former_v2 are shared in

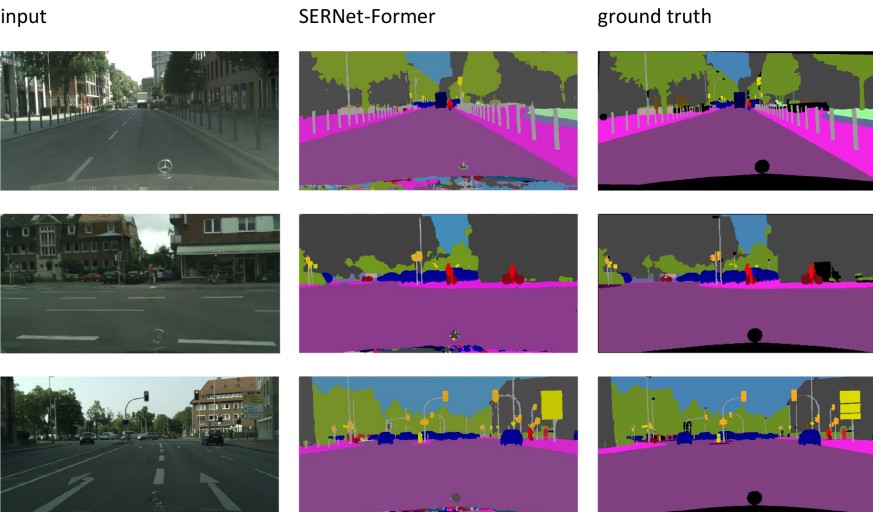

input            SERNet-Former          ground truth

Figure 3: Examples from the inference results on Cityscapes *validation* dataset.

Tables 2, 3, and 4. The implementation details of these selected state-of-the-art networks are shared in the Appendix.

### 4.3 EVALUATION OF THE PERFORMANCE OF THE NETWORK WITH APPLIED METHODS

In assessing the performance of the developed network, SERNet-Former, training accuracies as well as loss are observed, besides mean IoU, at the initial stages of training (Fig. 1). SERNet-Former demonstrated state-of-the-art performance on CamVid, significantly improving accuracy in identifying small objects (e.g., poles and bicycles) and occluded objects (e.g., trees and fences) that are often misclassified as buildings (Table 1) (Erişen, 2024). The novel AbM and AfN modules effectively reduced loss in early training epochs, enhancing overall accuracy (Fig. 1) (Erişen, 2024). On Cityscapes, SERNet-Former excelled at recognizing distant objects and diverse classes, including sidewalks, vegetation, terrain, sky, and vehicles (e.g., cars, trucks, and trains), as well as dynamic entities like pedestrians and riders (Table 2, Fig. 3) (Erişen, 2024). The results demonstrate how effective the proposed methods are in recognizing complex scenes and urban environments.

The integration of attention-boosting gates (AbGs) and attention-boosting modules (AbMs) into InternImage-XL also yielded a measurable improvement in test performance on the Cityscapes dataset, increasing from 84.85 to 85.1. Such improvements are significant in real-world applications where even small performance boosts can have meaningful impacts. Furthermore, this achievement demonstrates the adaptability of AbG and AbM when applied to different architectures, reaffirming their potential as versatile tools for enhancing deep learning models.

AbGs, which are also deployed in AfNs, provide efficient transfer of matrix weights before the matrix multiplications and AbMs provide fusion by the concatenation after multiplication operations that improve the network performances without compromising the hardware performance. In following the efficiency of the initial performance of the networks throughout the applied methods, training accuracy and loss were also critical performance parameters besides mean IoU. The results also reveal that AbM and AfN decrease the loss in the initial training epochs successfully, increasing the actual test performance and accuracy of SERNet-Former (Fig. 1) (Erişen, 2024).

### 4.4 ABLATION STUDIES

Ablation studies are performed on the checkpoints of SERNet-Former_v2, trained and tested on the Cityscapes dataset in PyTorch. Since element-wise additions and concatenations fuse modules of AbM and AfN, ablation studies are performed by removing the added methods and validating the

Table 5: Ablation studies on the Cityscapes testset

| $AbM_5$ | $AfN_1$ | $AfN_2$ | $DbN$ | mIoU |
|---|---|---|---|---|
| | | | | 68.71 (-15.72) |
| ✓ | ✓ | | ✓ | 77.04 (-7.39) |
| ✓ | | ✓ | ✓ | 80.2 (-4.23) |
| ✓ | ✓ | ✓ | | 82.6 (-1.83) |
| | ✓ | ✓ | ✓ | 84.04 (-0.39) |
| ✓ | ✓ | ✓ | ✓ | 84.43 |

results using checkpoints (Table 5). DbN is replaced with a simpler convolutional network without dilation factors, and the results are also reported in Table 5.

$AbM_5$ is chosen for the ablation studies because $AbM_4$, $AbM_3$, and $AbM_2$ significantly disrupt the network's stability when removed. Therefore, $AbM_5$ is the most reliable version for evaluating the influence of attention-boosting gates. In this context, the attention-boosting module ($AbM_5$), added to the final convolution block of the encoder, enhances the feature-based and pixel-wise contextual semantic information of SERNet-Former_v2 by 0.39 percent (Table 5). The DbN module further improves the network, contributing to a 1.83 percent performance increase over the baseline (Table 5).

AfNs are integrated into the deconvolution layers with different strides in the decoder of SERNet-Former_v2. When AfN is introduced into the deconvolution layer with stride 1 ($AfN_1$), it processes the global semantic information, resulting in a 4.23 percent improvement in overall test performance (Table 5). When AfN is fused into the deconvolution layer with stride 4 ($AfN_2$), the network performance improves by 7.39 percent, demonstrating the contribution of the method in handling local semantic information (Table 5). Thus, AfNs significantly enhance segmentation networks by deploying attention-based algorithms to process spatial information across different scales and contexts in the decoder.

## 5 CONCLUSION

Attention-boosting gates and modules are utilized in advanced segmentation networks like SERNet-Former and InternImage to address the multi-scale challenge of integrating semantic information across varying sizes and contexts. The Sigmoid function helps to activate the pixel-wise feature maps. Respectively, attention-fusion networks improve the performance and efficiency of the decoder of SERNet-Former and SERNet-Former_v2 in fusing the semantic data from different contexts significantly.

Limitations: It is found that SERNet-Former architectures are still the most compatible networks for implementing the attention-based methods of Abg, AbM, and AfN. On the other hand, our methods are also limited to the overall architecture of SERNet-Former and InternImage in this article. For instance, it is not possible to deploy AfNs to InternImage as its architecture has already exploited the attention-based transformers throughout the whole network. Nevertheless, it is apparent that the methods that we apply, but not limited to SERNet-Former and InternImage-XL, can contribute to the results of different baselines and even state-of-the-art segmentation networks without compromising computational performance.

Future work: It is also possible to deploy AfNs in the decoder of networks. Adding multi-head attention networks, larger transformer heads, and adapting the groups of attention-fusion modules to ViT baselines can also improve the networks' results. Our attention-based modules can also contribute to the activations of learning representations in the baselines to be deployed in video and action recognition. Respectively, we hope that our methods deployed in Efficient-ResNet, InternImage-XL, SERNet-Former, and SERNet-Former_v2 inspire many researchers to develop novel and efficient state-of-the-art networks and applications in integrating CNN with novel transformer mechanisms and attention-based algorithms.

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

# A  APPENDIX

## A.1  ADDITIONAL METHODS: DILATION-BASED SEPARABLE CONVOLUTION NETWORKS

Dilation-based networks (DbNs) decompose outputs into finer feature maps, enriching semantic fusion between the encoder and decoder stages (Fig. 2) (Chen et al., 2018; Erişen, 2024). In DbN, the output of the encoder is processed and transferred into the convolutional layers with the dilation factors of 1, 12, 16, and 18 that are followed by the batch normalization and ReLU layers. The outputs of activation layers in DbN are fused before the decoder.

## A.2  ADDITIONAL METHODS: LOSS FUNCTION AND THE CLASSIFICATION LAYER

For calculating the performance of semantic segmentation networks, the cross-entropy loss function is deployed in Equation (5).

$$loss = -\sum_{x \in classes}^{C} T(x) \times \log(Y(x)), \tag{5}$$

where $T$ denotes the target, $x$ is a class in the labeled classes $C$ in a dataset (Erişen, 2024). $Y$ represent the predicted pixels. Prior to experimentation, class weights for each dataset are calculated and integrated into the pixel classification layer. This ensures balanced training across all classes. The cross-entropy loss function is used (Equation 5) to compute the difference between the networks' predictions and the ground truth labels, facilitating efficient network optimization.

## A.3  IMPLEMENTATION DETAILS OF SERNET-FORMER_v2 AND INTERNIMAGE-XL IN PYTORCH

SERNet-Former_v2 is recompiled in PyTorch depending on the methods that are described in developing SERNet-Former. It is trained and tested on the Cityscapes, ADE20K, and BDD100K, which are popular benchmarking datasets for training and testing segmentation networks on urban and scene-level understanding tasks. In the development of SERNet-Former_v2 in PyTorch, attention-based algorithms of dense, fully connected forward predictions are also applied in the re-implementation of AbGs, AbMs, and AfNs, resulting in a larger and yet much more robust network with efficient results on the Cityscapes testset as well as ADE20K and BDD100K datasets (Table 4). InternImage-XL is also modified by applying the methods of AbGs and AbMs to the activation function, and the network is trained and tested on the Cityscapes dataset.

Table 6: State-of-the-art test results on the CamVid dataset

| Method | mIoU | Baseline Architecture | Params (M) |
|---|---|---|---|
| VideoGCRF | 75.2 | ResNet-101 | - |
| CCNet | 79.1 | ResNet-101 | - |
| Zhu et al. (2019) | 82.9 | WideResNet38 | - |
| RTFormer-Slim | 81.4 | RTFormer blocks | 4.8 |
| RTFormer-Base | 82.5 | RTFormer blocks | 16.8 |
| SegFormer B5 | 83.7 | MiT-B5 (IM-1K, MV) | 84.7 |
| **SERNet-Former** | **84.6** | **Efficient-ResNet** | 44.2 |

**Cityscapes:** SERNet-Former_v2 is re-compiled in the PyTorch platform and initially trained on the Cityscapes dataset for 160k iterations with the crop sizes of 1024 by 1024. The learning rate was set to 1e-5 using SGD optimizer with momentum 0.9 and the changing l2r regularization values are used throughout the training schedules. The results are submitted to the official evaluation server, which returns 85.02 mIoU (Table 4).

InternImage-XL, which is pre-trained first on Mapillary Vistas (Neuhold et al., 2017) and the Cityscapes dataset for 160K iterations by using Upernet baseline with the crop sizes of 512 by 1024 (Wang et al., 2023), is first tested on the Cityscapes dataset (Table 4). Then, the pre-trained model is modified by applying attention-boosting methods to its activation function. The network is trained for 80K iterations and tested on the Cityscapes dataset by submitting the results to the evaluation server. The results have shown that attention-boosting modules improved InternImage-XL's performance by 0.25 mIoU on the Cityscapes test dataset (Tables 2, 3, 4).

**ADE20K Dataset:** SERNet-Former_v2 is trained on ADE20K training dataset for 160k iterations with the learning rate of 1e-6 using SGD optimizer with momentum 0.9 and the changing l2r regularization values by using crop sizes of 896 by 896, based on the training dataset with 150 classes, which are the same as those in the Coco-Stuff 164k dataset. The network is evaluated through the validation dataset and returns 59.35 mIoU (Table 4).

**BDD100K:** SERNet-Former_v2, trained on the Cityscapes dataset, is applied directly to the BDD100K validation dataset for semantic segmentation, which contains 10000 images and 19 annotated classes, the same as those in Cityscapes. The network sets state-of-the-art results on the BDD100K validation dataset, 67.42 mIoU (Table 4).

The networks are trained and tested using NVIDIA L4 with 24 GB GPU memory in PyTorch environments.

A.4 ADDITIONAL RESULTS AND ILLUSTRATIONS ON THE SELECTED DATASETS

Table 6 and Fig. 4 illustrate the performance of SERNet-Former on the CamVid test dataset with 233 images. Similarly, the colored illustrations of the prediction results of SERNet-Former and SERNet-Former_v2 on the Cityscapes test dataset are shared in Fig. 5.

In Fig. 6, the prediction results of SERNet-Former_v2 on ADE20K validation dataset with 2000 images and 150 annotated classes are presented. Fig. 7 illustrates the output of SERNet-Former_v2 network, trained on the Cityscapes dataset and directly evaluated on the BDD100K validation dataset with 10000 images represented through 19 annotated semantic classes sharing the same palette of labelIDs with the Cityscapes dataset.

input                    SERNet-Former                    ground truth

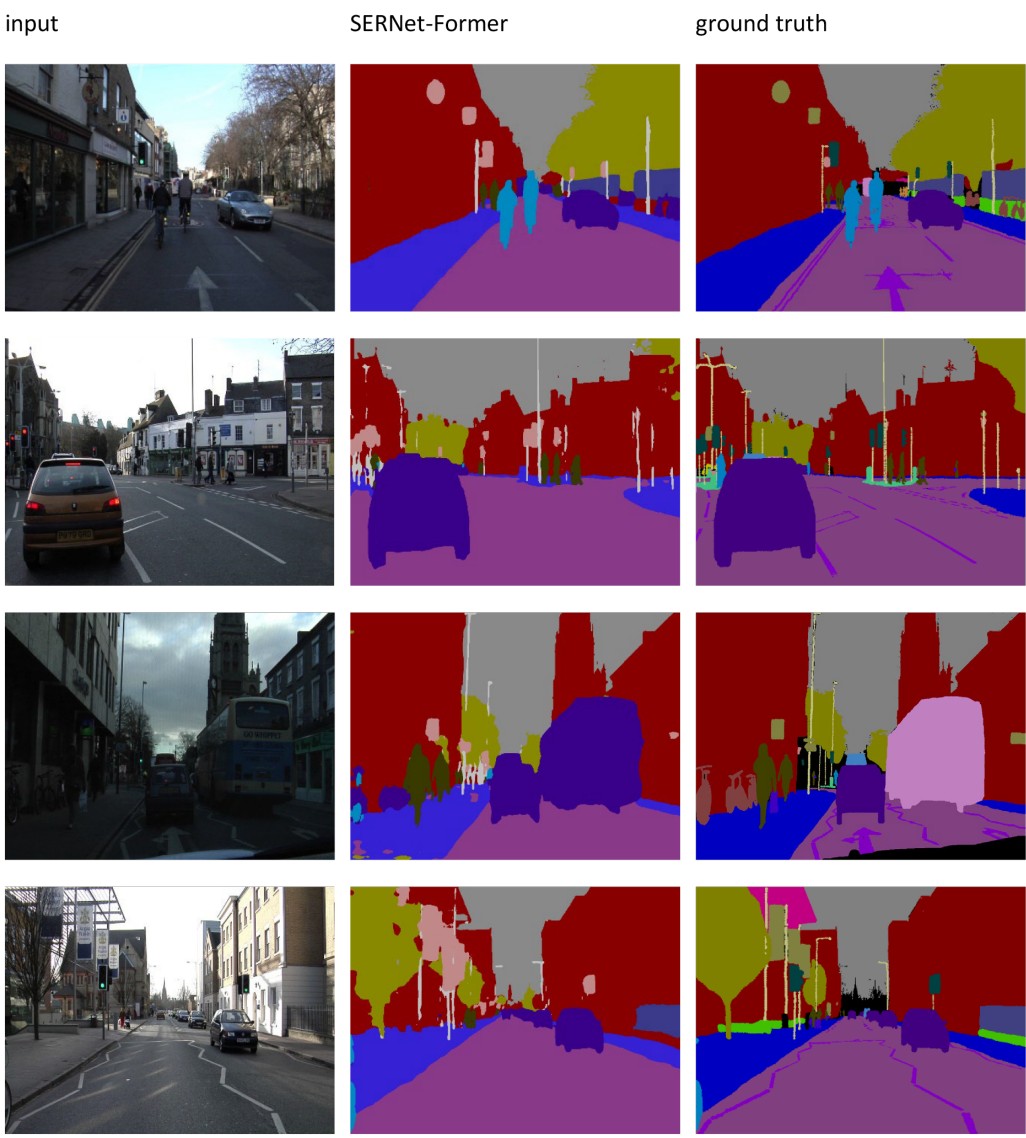

Figure 4: Segmentation results of SERNet-Former on the CamVid test dataset. Left column: Image inputs. Middle column: Prediction outputs of SERNet-Former. Right column: Ground truth of annotated labels.

input    SERNet-Former    SERNet-Former_v2

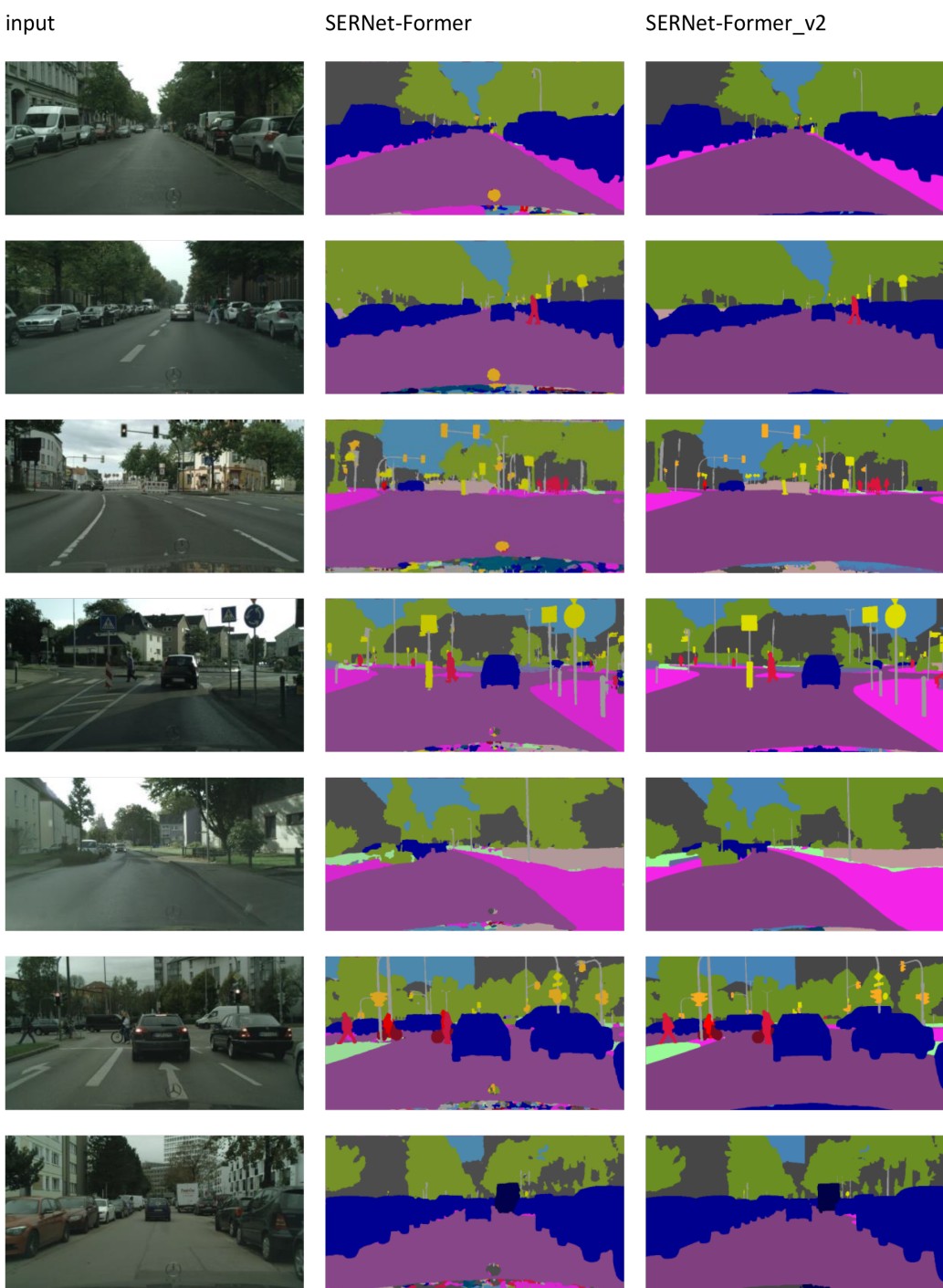

Figure 5: Segmentation results of SERNet-Former and SERNet-Former_v2 on Cityscapes test dataset. Left column: Image inputs. Middle column: Prediction results of SERNet-Former. Right column: Prediction results of SERNet-Former_v2.

input           SERNet-Former_v2         ground truth

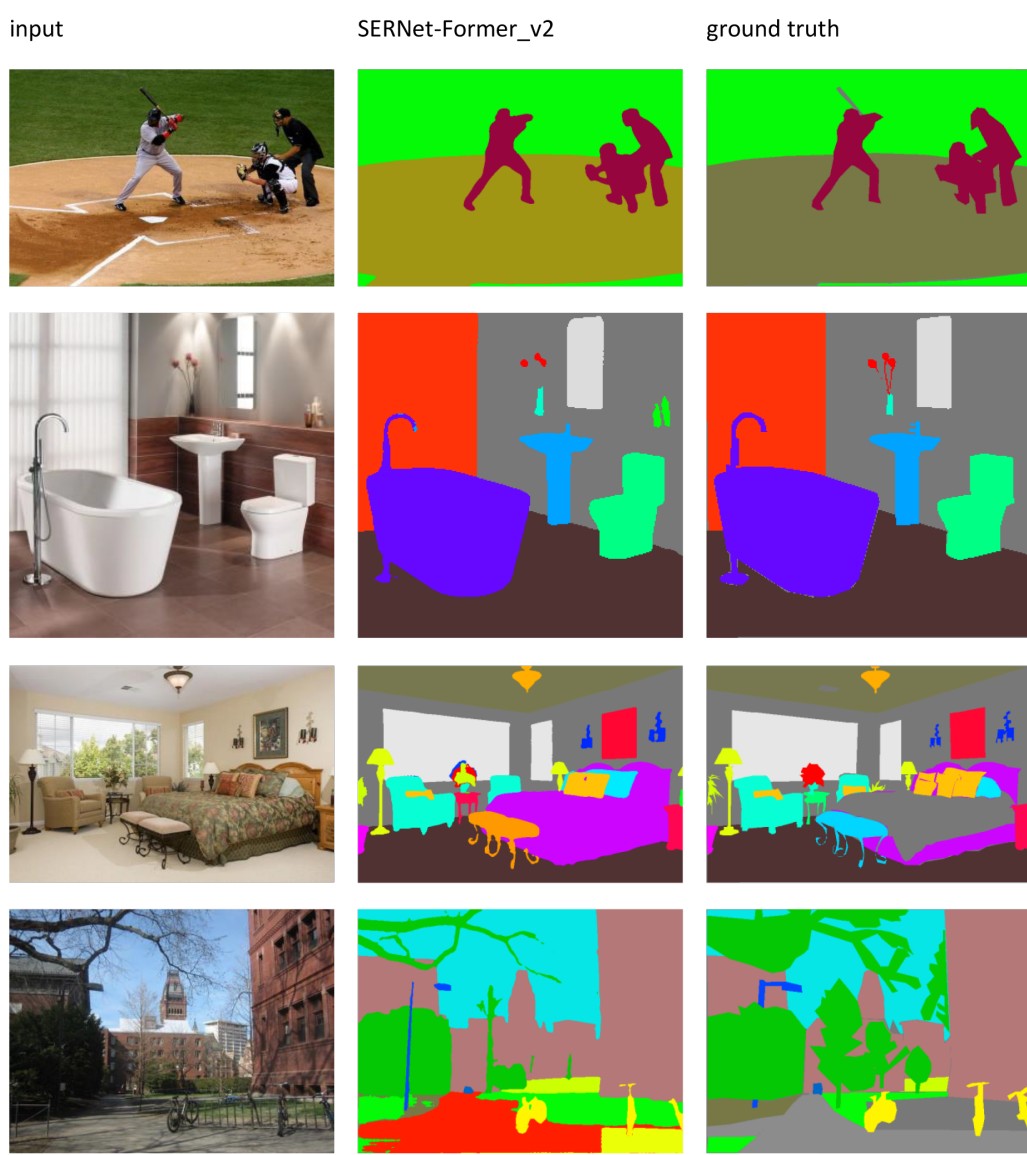

Figure 6: Segmentation results of SERNet-Former_v2 on ADE20K validation dataset. Left column: Image inputs. Middle column: Prediction results of SERNet-Former_v2. Right column: The ground truth of annotations.

input          SERNet-Former_v2          ground truth

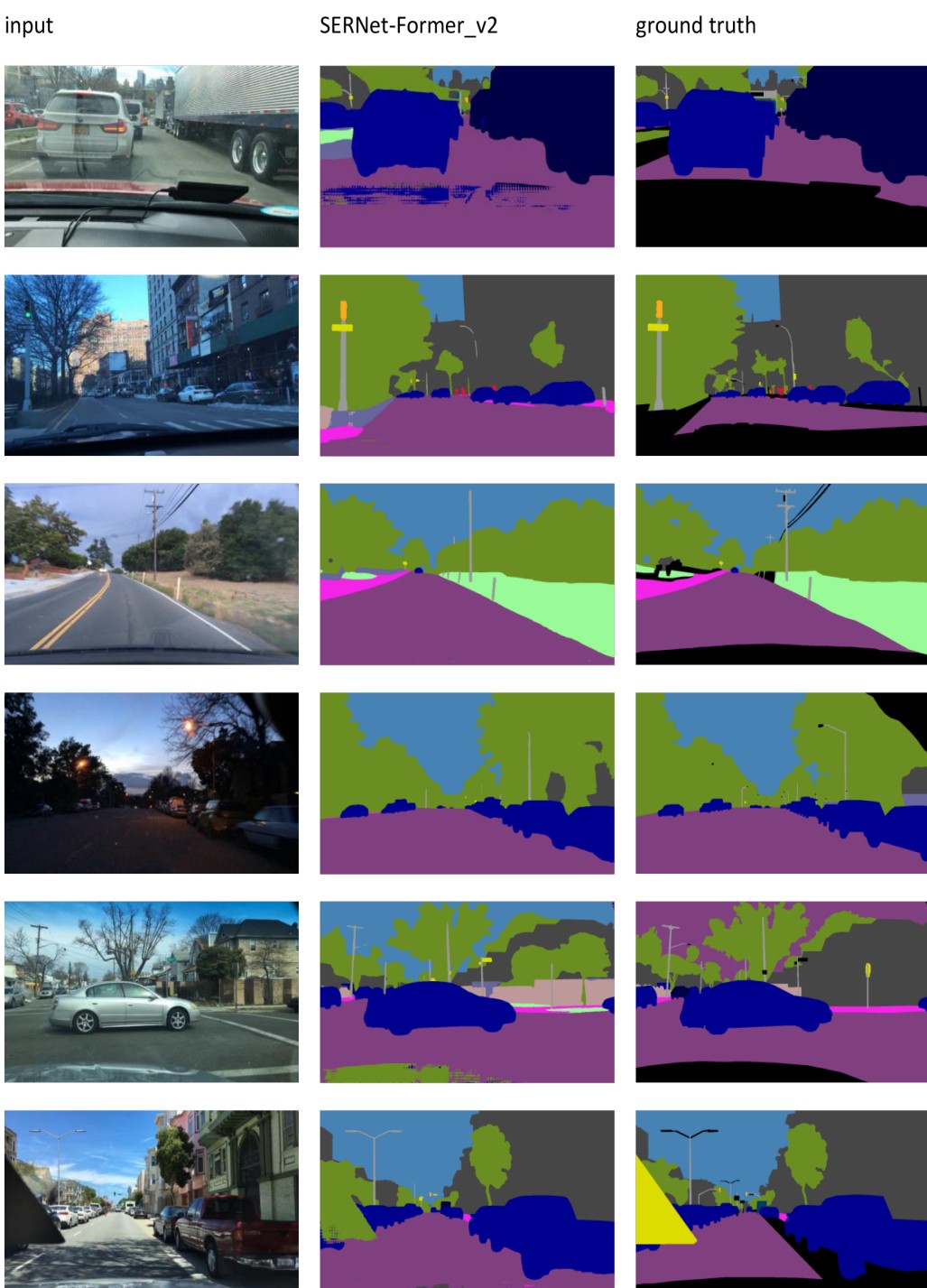

Figure 7: Segmentation results of SERNet-Former_v2 on BDDlOOK validation dataset. Left column: Image inputs. Middle column: Prediction results of SERNet-Former_v2. Right column: The ground truth of annotations.

