# OpenReview forum: "Segmentation using efficient residual networks with attention-fusion modules"
_ICLR.cc/2025/Conference — ICLR 2025 Conference Withdrawn Submission_

### Official Review · Reviewer_sF1B · 2024-10-30

**Soundness:** 2
**Presentation:** 1
**Contribution:** 2
**Rating:** 5
**Confidence:** 4

**Summary:**

This paper proposes AbGs,AfNs to fuse global and local semantic information in segmentation. Attention-fusion networks are desined in the decoder part to improve the efficiency.

**Strengths:**

1.This paper has made useful explorations in the fusion of global and local information, bringing some inspiration to this field.
2.Experimental results show that this method has some advantages

**Weaknesses:**

1.The writing of this article is poor. Many sentences are not clear and not easy to understand. Some sentences are too long and difficult to understand. e.g. line 125: The multi-scale problem in computer vision can be described as the discrepancy in integrating the different sizes of spatial and channel-based semantic information of an object acquired from the global and local contexts of segmentation networks.
line 203：It is aimed at developing an encoder-decoder architecture with additional attention mechanisms to get efficient segmentation networks fusing semantic information from different contexts by regarding the multi-scale problem.

2.The method in this article lacks insight, and many designs are tricky.  e.g. Why are there two consecutive layers (AbM4, AbM5) in H/8 and W/8 resolutions? For another question, please refer to Question 2.

3.The experimental analysis is not enough, and the ablation experiment is not very sufficient. More details can refer to Question 4.

4.This paper seems to have multiple submissions，which was accepted by IEEE CVMI previously.

**Questions:**

1. Why is the sigmoid function used as the activation function in the AbG module? Will it aggravate the gradient vanishing problem during training? Have you tried other activation functions?
2. Dilation-based convolution is used in DbN module, why not use dilation convolution in encoder and decoder part?
3. During upsampling, the image size changes from H/4, W/4 to H, W. Why not use progressive upsampling?
4. From Table 3 and Table 4, your performances are not as good as InternImage and VitAdapter-L(test mIoU). What are the parameters and inference speed(e.g millisecond) of these two methods? How do they compare to yours?

---

> ### Author Response · Authors · 2024-11-17
> **Comments to the reviewer sF1B's evaluations about the weaknesses of the paper**
>
> Thank you for your comments and evaluations.
>
> **W1**. The entire text has been revised per your suggestions. Examples include:
>
> "The multi-scale problem in computer vision refers to the challenge of integrating spatial and channel-based semantic information of an object in segmentation networks from both global and local contexts \cite{r64}."
>
> "This research aims to assess the attention-based mechanisms of attention-boosting gates (AbGs), modules (AbMs), and attention-fusion networks (AfNs) in improving efficient segmentation networks."
>
> **W2**. Based on prior experiments, AbM4 causes abrupt changes in results when removed, making AbM5 the most suitable for ablation studies. Removing modules reduces the network's performance and stability, to be presented again in the revised ablation studies.
>
> **W3**. Ablation studies on the Cityscapes test dataset confirm results consistent with earlier submissions. The ablation section has been rewritten to reflect this with new results that are announced in Table 5 as well:
>
> Table 5: **Ablation studies** on the Cityscapes testset
>
> | AbM5 | AfN1 | AfN2 | DbN | mIoU |
> | --- | --- | --- | --- | --- |
> |  |  |  |  | 68.71 (-15.72) |
> | ✓ | ✓ |  | ✓ | 77.04 (-7.39) |
> | ✓ |  | ✓ | ✓ | 80.2 (-4.23) |
> | ✓ | ✓ | ✓ |  | 82.6 (-1.83) |
> |  | ✓ | ✓ | ✓ | 84.04 (-0.39) |
> | ✓ | ✓ | ✓ | ✓ | 84.43 |
>
>
> 68.71 https://www.cityscapes-dataset.com/anonymous-results/?id=15080fd2b6f19fcaafb2fb9daba365fc5588b68e47b4e53fa38fc111ebedb1f1
>
> 77.04_https://www.cityscapes-dataset.com/anonymous-results/?id=adc1e822e01581145b12af68ab1e6b7d6e897129543bc298358a4a22fdeca85
>
> 82.60_https://www.cityscapes-dataset.com/anonymous-results/?id=faaa1bfe4a3da14f74dee88ae93e50db49f312160ba78d5ff0bc054576f036be
>
> 84.04 https://www.cityscapes-dataset.com/anonymous-results/?id=8728cc523c3779277805e758b3f2bd936de04372ece989a05e5ec29628e8e3bb
>
> 84.43 https://www.cityscapes-dataset.com/anonymous-results/?id=44b549bfc1e7bc433b3b32f18481f924208172f7d0bfed0c9274530430475034
>
> **W4**. We ensured that this work is distinct from submissions to CMVI 2024 or CVPRW 2024 and not under consideration elsewhere, including CVPR 2025. Please let us remind that, if there would be any overlapping content for publications, these works should have already been desk-rejected. In that regard, we believe that there would not be any issues about the submission and revision of this paper against the ICLR 2025 standards the evaluation committee.
>
> We have investigated the potential implications of earlier works and confirmed that the paper accepted at CVMI might discuss **the earlier version of SERNet-Former rather than the methods**. The updated versions, or newly conducted experiments we have presented cannot be presented. Moreover, it is important to note that **CVMI 2024** only accepts papers **limited to four pages**. Consequently, the accepted work diverges from the ArXiv version. It will likely omit substantial information, guaranteeing it misses at least half of the key points, methods, experiments, and implementation details outlined in our ArXiv paper.
>
> Additionally, **we can affirm that we finalized our experiments and updated our studies after the CVMI conference concluded**, **ensuring no overlap in content**. In our submission, we have **revised the methods, introduced new equations, and corrected numerous details from the referenced works**. In response to the inquiries from other reviewers regarding design and implementation details, we have included **several key points in this submission and enhanced the text to ensure that it is presented uniquely**.
>
> While some approaches in presenting original networks were overlapping, we revised the paper, enhanced the methods, applied them in PyTorch, and tested them on new networks. Thus, this ICLR 2025 submission/revision would have been entirely new, with proper citations and no overlapping content aside from explanatory context about segmentation and selected network features.

---

> ### Author Response · Authors · 2024-11-22
> **Comments to the reviewer sF1B's evaluations about the questions about the paper**
>
> **Q1**. Skip connections between AbGs, AbMs, and residual layers prevent gradient vanishing. ReLU combined with AbMs enhances this effect. From our earlier experiments, the Swish function was also experimented instead of ReLU in ResNet architectures, yet did not yield better results. Compared to ReLU, AbG with Sigmoid function outperforms other well-known activation functions, and the majority of the reasons are discussed in the paper based on attention-based algorithms and processes. Please also find the revised parts, not limited to this example, in the text in this revision:
>
> "… skip connections are designed for efficient multi-scale feature fusion in SERNet-Former's decoder, preventing gradient vanishing of the Sigmoid function applied via AbMs."
>
> Similar methods from our research are also applied to different networks using ReLU, GeLu, etc., such as InternImage. Of course, there can be many other approaches that we have not discovered yet. Nevertheless, we are consistent at least in what we have implemented.
>
> **Q2**. From prior experience, dilation-based convolution layers in the encoder with different strides and larger image sizes do not contribute to the network and increase its size. This may be caused by worsening the global semantic information. Thus, the authors did not attempt to add dilation factors to the convolution layers in the decoder. However, the same could be observed in the decoder as the global semantic information would dilute and the computational cost increase.
>
> **Q3**. From our earlier experience, the mentioned details of ladder-shaped decoder for progressive up-sampling with different stride factors were also tried, and the network ran for a while in our earlier design experiments of SERNet-Former. However, the results were much lower. The articles about in-depth surveys of DeepLab and HRNet architectures (Chen et al., 2018; Chen et al., 2020) also mention why the stride=4 returns better solutions. We also cited them in our paper.
>
> **Q4**. In this revision, we recompiled SERNet-Former in PyTorch for new results on Cityscapes, ADE20K, and BDD100K. As a nice coincidence, we were also trying to improve InternImage by applying our methods mentioned in the text, achieving improved Cityscapes test results (mIoU from 84.85 to 85.1):
>
> Table 2. Per-class accuracies (mIoU) based on Cityscapes test dataset
>
> | Methods | Road | Sidewalk | Building | Wall | Fence | Pole | Traffic Light | Traffic Sign | Vegetation | Terrain | Sky  | Person   | Rider    | Car | Truck | Bus  | Train         | Motorcycle   | Bicycle    |   mIoU      |
> | --- | --- | --- | --- | --- | --- | --- | --- | --- | --- | --- | --- | --- | --- | --- | --- | --- | --- | --- | --- | --- |
> | SERNet-Former† | 98.2 | 90.2 | 94.0 | 67.6 | 68.2 | 73.6 | 78.2 | 82.1 | 94.6 | 75.9 | 96.9 | 90.0 | 77.7 | 96.9 | 86.1 | 93.9 | 91.7 | 70.0 | 82.9 | 84.8 |
> | **SERNet-Former_v2††** (ours) | 98.8 | 88.3 | 94.6 | 72.6 | 69.5 | 73.3 | 78.7 | 83.2 | 94.2 | 74.7 | 96.2 | 88.9 | 76.6 | 96.5 | 84.3 | 95.3 | 92.7 | 77.0 | 79.8 | **85.0** |
> | --- | --- | --- | --- | --- | --- | --- | --- | --- | --- | --- | --- | --- | --- | --- | --- | --- | --- | --- | --- | --- |
> | InternImage-XL | 98.9 | 88.7 | 94.7 | 72.1 | 70.3 | 73.4 | 79.1 | 83.5 | 94.3 | 74.5 | 96.1 | 88.9 | 76.1 | 96.7 | 84.2 | 94.7 | 91.1 | 75.0 | 79.8 | 84.8 |
> | **InternImage-XL†††** (ours) | 98.9 | 88.7 | 94.7 | 72.8 | 70.2 | 73.4 | 79.1 | 83.5 | 94.3 | 74.5 | 96.2 | 88.9 | 76.2 | 96.7 | 85.0 | 95.2 | 92.4 | 76.2 | 79.9 | **85.1** |
>
> …
>
>
> Table 3. Results of state-of-the-art methods on Cityscapes datasets
>
> | Reference | Method | Baseline | test mIoU | validation mIoU |
> | --- | --- | --- | --- | --- |
> | CVMI 2024 | SERNet-Former | Efficient-ResNet | 84.8 | 87.35 |
> |  | **SERNet-Former_v2** (ours) | Efficient-ResNet v2 (ours) | **85.02** | 86.5 |
> | --- | --- | --- | --- | --- |
> | CVPR 2023 | InternImage-XL | UperNet | 84.85 | 86.2 |
> |  | **InternImage-XL** (ours) | UperNet | **85.1** | 86.5 |
>
>
> Table 4: Performance results of models developed and tested in PyTorch
>
> | Dataset | Model | mIoU | inference time (s/task) | parameters (M) |
> | --- | --- | --- | --- | --- |
> | ADE20K (2K validation) |  |  |  |  |
> |  | SERNet-Former v2 (ours) | 59.35 | 0.75 | 245 |
> | BDD100K (10K validation) |  |  |  |  |
> |  | SERNet-Former v2 (ours) | 67.42 | 0.75 | 245 |
> | Cityscapes (test) |  |  |  |  |
> |  | SERNet-Former v2 (ours) | 85.02 | 0.75 | **245** |
> |  | InternImage-XL (ours) | 85.10 | 0.76 | **368** |
>
> •	84.85: https://www.cityscapes-dataset.com/anonymous-results/?id=12cd5268fab7c1e7c21f2fbd82ed01a86c25e5b5fbd7dbec9782c93c5cb52a4b
>
> •	85.1: https://www.cityscapes-dataset.com/anonymous-results/?id=330be9f0cfb2f2dfce28fc62e52c4bdc83684fe006e6e8212599be63a03ceb7f
>
> You may find the original implementation details about these networks in The Appendix as well.
>
> We hope that the current revisions will meet your and other reviewers' expectations and you accept the revised version.

---

### Official Review · Reviewer_UEkU · 2024-11-03

**Soundness:** 2
**Presentation:** 2
**Contribution:** 1
**Rating:** 1
**Confidence:** 5

**Summary:**

-

**Strengths:**

-

**Weaknesses:**

Considering that this paper has already been accepted by IEEE CVMI, I think it should be rejected.

**Questions:**

-

---

> ### Author Response · Authors · 2024-11-17
> **Comments to the reviewer UEkU's evaluations with revision notes to be submitted**
>
> As far as the author(s) know, blind peer-reviewing standards are deployed in ICLR 2025. However, the author(s) cannot be sure about how the reviewer(s) can justify that the paper submitted to ICLR can be the same as the mentioned works on SERNet-XFormer with CVPR 2024 Workshops and CMVI 2024 announced in a GitHub repository that we are aware of. If a paper is published in IEEE, then that might also be significantly different from the already announced public papers!
>
> Plus, no CPCI conference article or IEEE Xplore paper has been published yet about the versions of SERNet-Former networks. The network is announced publicly, and the authors believe it is worth consideration. Thousands of similar papers in IEEE Xplore focus on similar approaches with similar titles and aims and even with different versions. Based on the reviewers' arguments, it is also possible to argue that using ResNet, DeepLab, or ViT architectures can result in plagiarism. The methods in this paper have been adopted, with many corrections and improvements on the methods from the ones applied in SERNet-Former announced in ArXiv. So, we cite them in the paper in this revision. Thus, we also share the results from earlier works and would like to provide additional results in this revision. We also want to share our paper's original repository for ICLR 2025 if you can reconsider your review.
>
> The focus on methods and highlights has been revised. Only some introductory words were used, and everything was fine with them. However, the necessary sentences are completely rewritten. We have also been adopting our methods on ViT and CNN architectures, such as InternImage, which are available for development. We have yet to announce the improved the results of InternImage not to violate the ICLR reviewing rules.
>
> As a brief, tweaking the scale factor and changing the algorithms for the activation function of InternImage resulted in better initial performance of InternImage T or S and XL on the Cityscapes dataset, and we are now getting 85.1 mIoU on Cityscapes official test dataset with XL. Even so, the improvement of our methods in residual network architectures was much more satisfactory and deployable. Thus, we decided to announce the implementation of methods and results on SERNet-Former. However, we will also announce the results on InternImage with a slightly different application of our methods.
>
> In brief, we applied the mentioned methods of AbG and AfN with slight changes in our paper, and we have also now revised our work as the original contribution to SERNet-Former, which is now recompiled in PyTorch platform getting 85.02 on Cityscapes official test dataset (publicly announced – anonymous authors). In this revision, we were anticipating announcing the completely new results and revisions, changes and improvements in the methods, implementation details, as well as other results of SERNet-Former_v2 also getting results on ADE20K dataset 59.35 mIoU, and BDD100K with 10K validation dataset with 67.42 mIoU in the revision of our paper and the appendix.
>
> We hope you reconsider your decision and do not waste the great efforts put into this research.

---

> > ### Comment · Reviewer_UEkU · 2024-11-22
> >
> > Considering that AbG and AfN are the methods proposed at IEEE CVMI 2024, I kindly express that I disagree that the current version has any essential changes from the previously published version. The author's proposal to apply these modules to more architectures and more data scenarios, and to refactor the code using pytorch, is not enough to constitute a contribution that meets ICLR standards. I still maintain my original attitude towards this paper.

---

> > > ### Author Response · Authors · 2024-11-22
> > > **No publication about the methods**
> > >
> > > Dear reviewer,
> > >
> > > There is not any particular publication about these methods. However, I think you violate or overruled your position with a bias that this work is published.

---

> ### Author Response · Authors · 2024-11-22
> **Search results about the claims**
>
> Can you please provide us the reliable, official results that support your claim that
> this submission and especially this revised version of this paper is submitted elsewhere and published in its current form.
>
> We have conducted plagiarism search and the paper is 100 % original now!
> Please do not forget that if this paper is overlapping with the content of published works,
> it should already have been desk rejected.
>
> In that regard, you, indeed, have already violated the blind peer-reviewing rules and still insisted on your attitude.
>
> Moreover, we provided original implementation details and methods about the attention-based modules in this revision
> that are not published elsewhere.
> However, we concerned that you did not even have a look at our submission at all.
>
> However, it will not be fair to ruin somebody's work on purpose.
>
> Sincerely best

---

> ### Author Response · Authors · 2024-11-28
> **The notice and kind reminder about the clarifications of questions of reviewer UEkU**
>
> Dear reviewer,
>
> Please reconsider your rating based on our latest submissions and the clarifications regarding the original details and our contributions to the existing works that have not been published elsewhere. We understand you may have strong feelings about our previous interactions and the existing works, but we emphasize that the publication of this manuscript is incredibly important to us. Your re-evaluation and potential increase in rating would greatly impact our efforts and goals.
>
> Let us clarify once again:
>
> We are sincerely disappointed that our work has been labeled as duplicated, leading to its rejection, particularly in light of the announcement regarding the acceptance of a paper about SERNet-Former at a conference. We would like to share additional insights to provide more clarity on this issue.
>
> We have investigated the potential implications of earlier works and confirmed that the paper accepted at CVMI might discuss **the earlier version of SERNet-Former rather than the methods**. The updated versions, or newly conducted experiments we have presented cannot be presented. Moreover, it is important to note that **CVMI 2024** only accepts papers **limited to four pages**. Consequently, the accepted work diverges from the ArXiv version. It will likely omit substantial information, guaranteeing it misses at least half of the key points, methods, experiments, and implementation details outlined in our ArXiv paper.
>
> Additionally, **we can affirm that we finalized our experiments and updated our studies after the CVMI conference concluded**, **ensuring no overlap in content**. In our submission, we have **revised the methods, introduced new equations, and corrected numerous details from the referenced works**. In response to the inquiries from other reviewers regarding design and implementation details, we have included **several key points in this submission and enhanced the text to ensure that it is presented uniquely**.
>
>
>
> We understand and respect that you might enjoy to reject it. However, after such a great effort in revising and improving the article regarding your inquiries and questions, the decision of strong rejection means something else. Thus, we still believe and would like to warn you that you could increase your rating no matter what you are concerned about. We assure you that the details and methods have yet to be partially or fully published elsewhere.
>
> Sincerely best
>
> Author(s)

---

### Official Review · Reviewer_HFav · 2024-11-04

**Soundness:** 3
**Presentation:** 3
**Contribution:** 3
**Rating:** 5
**Confidence:** 4

**Summary:**

The paper "SERNet-Former: Segmentation by Efficient-ResNet with Attention-Boosting Gates and Attention-Fusion Networks" introduces an innovative segmentation framework that leverages advanced attention mechanisms within a robust network architecture. The authors provide comprehensive experimental results that validate their approach against existing methods, demonstrating its effectiveness across multiple datasets. However, the manuscript could be improved by addressing the limitations of the method, expanding the discussion of the ablation studies, and including a broader range of comparative benchmarks. Overall, this work represents a significant contribution to the field of image segmentation.

**However**, I noticed that this paper has already been accepted by IEEE CVMI 2024, titled "SERNet-Former: Segmentation by Efficient-ResNet with Attention-Boosting Gates and Attention-Fusion Networks." You can view the acceptance list for the conference at https://cvmi2024.iiita.ac.in/AcceptedPapers.php. After comparing the version in the GitHub repository (https://github.com/serdarch/SERNet-Former) with the version submitted by the authors to ICLR, it appears that there are only minimal differences between the two.

**Strengths:**

- **Innovative Approach**: The paper presents a novel method, SERNet-Former, which combines Efficient-ResNet with attention mechanisms, demonstrating a promising advancement in segmentation tasks.
- **Comprehensive Experiments**: The authors conduct extensive experiments across various datasets, showcasing the effectiveness of their approach and providing a thorough comparison with existing methods.
- **Clear Presentation**: The manuscript is well-organized, with a logical flow that makes the methodology and results easy to follow, enhancing the overall readability.

**Weaknesses:**

- **Limited Discussion on Limitations**: The paper could benefit from a more in-depth discussion of the limitations of the proposed method, particularly in relation to different types of data or specific segmentation challenges.
- **Insufficient Detail in Ablation Studies**: While the authors present some ablation studies, additional detail on the impact of each component in the network would strengthen the understanding of their contributions.
- **Comparative Analysis**: The comparison with state-of-the-art methods could be more robust, particularly by including more recent benchmarks to provide a clearer context for the performance claims.

**Questions:**

I noticed that this paper has already been accepted by IEEE CVMI 2024, titled "SERNet-Former: Segmentation by Efficient-ResNet with Attention-Boosting Gates and Attention-Fusion Networks."

Additionally, I have a question regarding:

1. **What specific metrics were used to evaluate the performance of SERNet-Former compared to existing segmentation methods, and how do these metrics support the claims made by the authors regarding its effectiveness?**

2. **Can the authors provide more details on the design choices behind the attention mechanisms used in SERNet-Former and how they contribute to the model's overall performance in segmentation tasks?**

---

> ### Author Response · Authors · 2024-11-17
> **Comments to the reviewer HFav's evaluations with revision notes to be submitted**
>
> Thanks for your evaluations.
>
> First, please let us inform that with the current revisions of this paper is not published or submitted elsewhere.
>
> **W1**.Limitations are discussed in the revised article. We share a portion of the text from the revisions:
>
> “**Limitations**: It is found that SERNet-Former architectures are still the most compatible networks for implementing the attention-based methods of Abg, AbM, and AfN. On the other hand, our methods are also limited to the overall architecture of SERNet-Former and InternImage in this article. For instance, it would not even be possible to deploy AfNs to InternImage as its architecture has already exploited the attention-based transformers throughout the whole network. Nevertheless, it is apparent that our methods that are applied, but not limited to SERNet-Former and InternImage-XL, can contribute to the results of different baselines and even state-of-the-art segmentation networks without compromising the computational performance.”
>
> **W2**.New ablation works are presented in the text based on the checkpoints of SERNet-Former_v2, a new implementation in PyTorch. You may find some official results on the Cityscapes dataset, which will be explained in detail again in the subsection of ablation studies.
>
> Table 5: **Ablation studies** on the Cityscapes testset
>
> | AbM5 | AfN1 | AfN2 | DbN | mIoU |
> | --- | --- | --- | --- | --- |
> |  |  |  |  | 68.71 (-15.72) |
> | ✓ | ✓ |  | ✓ | 77.04 (-7.39) |
> | ✓ |  | ✓ | ✓ | 80.2 (-4.23) |
> | ✓ | ✓ | ✓ |  | 82.6 (-1.83) |
> |  | ✓ | ✓ | ✓ | 84.04 (-0.39) |
> | ✓ | ✓ | ✓ | ✓ | 84.43 |
>
>
> 68.71 https://www.cityscapes-dataset.com/anonymous-results/?id=15080fd2b6f19fcaafb2fb9daba365fc5588b68e47b4e53fa38fc111ebedb1f1
>
> 77.04 https://www.cityscapes-dataset.com/anonymous-results/?id=adc1e822e01581145b12af68ab1e6b7d6ee897129543bc298358a4a22fdeca85
>
> 82.60 https://www.cityscapes-dataset.com/anonymous-results/?id=faaa1bfe4a3da14f74dee88ae93e50db49f312160ba78d5ff0bc054576f036be
>
> 84.04 https://www.cityscapes-dataset.com/anonymous-results/?id=8728cc523c3779277805e758b3f2bd936de04372ece989a05e5ec29628e8e3bb
>
> 84.43 https://www.cityscapes-dataset.com/anonymous-results/?id=44b549bfc1e7bc433b3b32f18481f924208172f7d0bfed0c9274530430475034
>
> **W3**. Abstract and the main text is revised. We share some sentences from the abstract:
>
> “…In this research, SERNet-Former is deployed on the challenging benchmarking datasets such as ADE20K, BDD100K, CamVid, and Cityscapes depending on the attention-based methods with new implementations of the network, SERNet-Former_v2. Our methods have also been implemented for InternImage-XL and have improved the performance of the network on the Cityscapes test dataset (85.1% mean IoU). Respectively, the results of the selected networks developed by our methods on the challenging benchmarking datasets are found worth considering: 85.1% mean IoU on the Cityscapes test dataset, 59.35% mean IoU on the ADE20K validation dataset, 67.42% mean IoU on BDD100K validation dataset, and 84.62% mean IoU on the CamVid dataset.”
>
> **Table 4: Performance results of models developed and tested in PyTorch**
>
> | Dataset | Model | mIoU | inference time (s/task) | parameters (M) |
> | --- | --- | --- | --- | --- |
> | ADE20K (2K validation) |  |  |  |  |
> |  | SERNet-Former v2 (ours) | 59.35 | 0.75 | 245 |
> | BDD100K (10K validation) |  |  |  |  |
> |  | SERNet-Former v2 (ours) | 67.42 | 0.75 | 245 |
> | Cityscapes (test) |  |  |  |  |
> |  | SERNet-Former v2 (ours) | 85.02 | 0.75 | **245** |
> |  | InternImage-XL (ours) | 85.10 | 0.76 | **368** |
>
> 85.1 https://www.cityscapes-dataset.com/anonymous-results/?id=330be9f0cfb2f2dfce28fc62e52c4bdc83684fe006e6e8212599be63a03ceb7f
>
> *to be continued*

---

> ### Author Response · Authors · 2024-11-26
> **Thanks for your evaluation and hope**
>
> Many thanks for your evaluation.
>
> Indeed, we have put a lot of efforts on the revisions and we were expecting to be accepted as it currently stands as not enough to be accepted.
>
> We would like to improve text if you want to provide further revisions.

---

> ### Author Response · Authors · 2024-11-28
> **Edited Official comments to the reviewer HFav**
>
> *edited and continued*
>
> **Q1&2**.The specific metrics and design tricks/choices followed in developing attention mechanisms are mentioned in many places in the text. Here you may find some examples from the revision:
>
> “In assessing the performance of the developed network, SERNet-Former, training accuracies as well as loss are observed besides mean IoU, at the initial stages of training (Fig. 1).”
> …
>
> “AbGs, which are also deployed in AfNs, provide efficient transfer of matrix weights before the matrix multiplications, and AbMs provide fusion by the concatenation after multiplication operation, improving the network performances without compromising the hardware performance. In following the efficiency of the applied methods on the initial performance of the networks, training accuracy and loss were also critical performance parameters besides mean IoU. The results also reveal that AbM and AfN decrease the loss in the initial training epochs successfully, increasing the actual test performance and accuracy of SERNet-Former (Fig.1) \cite{r64}.”
>
> And, here is one significant detail forgotten but have revised in the text:
>
> “AbGs are initially designed alongside attention mechanisms and then modified into the weightless mathematical operators to be adapted to networks deploying attention transformers.”
>
> Thus, this revision includes many changes and additions, in addition to the examples presented here, to answer your inquiries.
>
> We would like to get your questions and comments, if any, about the recent submission as well.
>
> We anticipate that you will increase your rating and accept our recent submission.

---

### Official Review · Reviewer_UNaU · 2024-11-08

**Soundness:** 2
**Presentation:** 2
**Contribution:** 2
**Rating:** 3
**Confidence:** 4

**Summary:**

This paper introduces a transformer with an encoder-decoder structure to fuse the global and local information from the image for semantic segmentation. The proposed method improves the semantic segmentation on multiple datasets, demonstrating its effectiveness.

**Strengths:**

The proposed method improves the segmentation results based on the transformer architecture, which is lightweight.

**Weaknesses:**

1. It should be noted that IEEE CVMI has accepted the manuscript "SERNet-Former: Segmentation by Efficient-ResNet with Attention-Boosting Gates and Attention-Fusion Networks." Although CVMI has not yet provided the official version of the accepted paper, the author has provided a GitHub repository, which indicates that CVMI has accepted the paper. Furthermore, the figures and experimental results in the GitHub repository with arXiv and CVPRW versions are the same as those in the paper submitted to ICLR. The author should clarify this.

2. Apart from the above point, I find that this paper's presentation is of low quality. It lacks the motivation to propose a new method of fusing local and global semantics, which has been well-known for improving semantic segmentation performance. I suppose this motivation is presented in the introduction, which is missed in every part of the paper. Though the performances have been compared in the experimental section, I still cannot figure out why the proposed method yields better results. Furthermore, the presentation of the method lacks the necessary information. The critical Figure 2 fails to provide a clear illustration of the method. The relationship between Figure 2 and the equations is also unclear. This fact further disallows the reader to understand the insight behind the method.

3. Though the proposed method improves the segmentation results, it still lags behind other methods on important datasets (see test set on Cityscapes in Tab 4).

Based on the above points, I believe this submission fails to meet the ICLR standard and recommend its rejection.

**Questions:**

See the "Weaknesses" above.

---

> ### Author Response · Authors · 2024-11-17
> **Comments to the reviewer UNaU's evaluations with revision notes to be submitted**
>
> Thanks for your evaluation.
>
> W1. First of all, let us clarify that this paper has been completely revised to report the efficiency of our attention-based methods applied to some networks and new datasets, yet not to acclaim a specific network or work. We ensured that this work cannot be the same as those accepted to CMVI 2024 or CVPRW 2024, and not published. We now cite other works in the paper properly with regard to the author guidelines. We also shared the results from earlier works and were anticipating providing additional results in this revision. We are also considering sharing our paper's original repository for ICLR 2025 if you mind accepting this submission. We have revised the whole text in this revision. Here are the examples from the title and partial-abstract:
>
> “Efficient segmentation using attention-fusion modules
>
> Fusing global and local semantic information in segmentation networks remains challenging due to computational costs and the need for effective long-range recognition. Based on the recent success of transformers and attention mechanisms, this research applied attention-based mechanisms such as attention-boosting modules and attention-fusion networks in enhancing the performance of state-of-the-art segmentation networks, such as InternImage and SERNet-Former, addressing these challenges. Integrating attention-boosting modules into residual networks enhances baseline architectures like Efficient-ResNet, enabling them to extract global context feature maps in the encoder while minimizing computational costs. Attention-based algorithms can also be applied to networks utilizing vision transformers and convolutional layers, such as InternImage, to improve the existing results of state-of-the-art networks. In this research, SERNet-Former is deployed on the challenging benchmarking datasets such as ADE20K, BDD100K, CamVid, and Cityscapes depending on the attention-based methods with new implementations of the network, SERNet-Former_v2. Our methods have also been implemented for InternImage-XL and improved the test performance of the network on the Cityscapes dataset (85.1% mIoU)..."
>
> Hence, we must admit that the paper submitted to ICLR 2025 is not currently submitted elsewhere, such as CVPR 2025! The paper submitted to ICLR 2025 is completely rewritten, and the related works adopted in this research, to test the selected methods, are properly cited. There is no overlapping content now, yet there were only some explanatory and complementary sentences about segmentation and the features of selected networks.
>
> W2. Thanks for your warning. We have recognized some missing details in Figure 2 about AbG, AbM, and AfN. We have now revised Figure 2 and the text. As discussed, SERNet-Former architecture is the most compatible network to show the details of attention-based mechanisms in Figure 2. We also tried InternImage in this revision, and many other networks are not to be mentioned. Thus, the applied methods are also limited to the earlier works you are concerned about, and we discuss them in the limitations section.
> Indeed, the methods deployed in AbGs are as clear as shown in equations and figures. The author(s) recognized that the reviewers might confuse AbGs with other attention mechanisms, such as self-attention. On the other hand, the inner processes of transforming inputs, resizing, or forwarding the inputs or keys have already been explained in paragraphs. Since they were not the original contributions of this work, they are not illustrated and they can be found in many other related works. Thus, it can be concluded that our methods are also designed to be integrated into other attention mechanisms and networks. For instance, AbGs in AfNs and in InternImage-XL. Explanations can also be found in the revised text.
>
> W3. Getting the top result is not the primary focus of this study. Instead, we have been focusing on how and to what extent our methods contribute to the existing baselines and pre-trained networks. We also experimented with new models, such as InternImage-XL, and improved its test results from 84.85 to 85.1. You may also find some official results that are not published elsewhere:
>
> 84.85 https://www.cityscapes-dataset.com/anonymous-results/?id=12cd5268fab7c1e7c21f2fbd82ed01a86c25e5b5fbd7dbec9782c93c5cb52a4b
>
> 85.1 https://www.cityscapes-dataset.com/anonymous-results/?id=330be9f0cfb2f2dfce28fc62e52c4bdc83684fe006e6e8212599be63a03ceb7f
>
> &
>
> SERNet-Former_v2 on Cityscapes test dataset= 85.02 https://www.cityscapes-dataset.com/anonymous-results/?id=330be9f0cfb2f2dfce28fc62e52c4bdc83684fe006e6e8212599be63a03ceb7f
>
> We revised Tables 3 and 4 and added Tables 5 and 6, providing new results on new/modified models SERNet-Former_v2 and InternImage-XL and datasets Cityscapes, ADE20K, BDD100MK, and ablation works on the Cityscapes test dataset.
> We hope this helps to meet your expectations and that the efforts we put into this revision help to change your decisions in a positive direction.

---

### Author Response · Authors · 2024-11-19
**Official Comments to the Reviewers about the Revision_1**

Dear reviewers,

Thanks for your comments and evaluations.

We would like to present the last condition of the revised draft that is improved according to your comments and questions.
First, let us clarify that this paper has been completely revised to report the efficiency of our attention-based methods applied to some networks and new datasets, yet not to acclaim a specific network or work. We have ensured that this work is distinct from the mentioned papers accepted at CVMI 2024 or CVPRW 2024, and apparently has not been published elsewhere. However, we have adopted many details from the state-of-the-art works that the reviewers mention by properly citing them throughout the whole text, completely rewritten according to this revision's changing aim and motivation. Moreover, we worked on new models to assess the efficiency of our methods applied to encoder-decoder architectures using CNNs and attention mechanisms such as SERNet-Former_v2 and InternImage-XL using datasets ADE20K, BDD100K, Cityscapes, CamVid, Mapillary Vistas. Accordingly, we shared many new results in Tables and we hope that you find this revision improved.

Briefly, we have revised the title, abstract, introduction, highlights, methods, experiments, tables and figures, ablations work, conclusions, and references sections regarding your comments to improve the article. We also add new details about the implementations of the networks and datasets in the appendices that we newly worked on.

Following the comments and questions, the authors recognized some missing details in Figure 2 and revised them as well as the narration of the motivation and methods. On the other hand, it has been decided to keep the SERNet-Former architecture, which is still the most compatible network, showing the influence of the applied methods on different parts of a network. We also recompile this architecture in PyTorch, called SERNet-Former_v2, which is not published elsewhere, and we present its implementation details and results in this revision. As discussed, this revision reports new datasets and application details. We also conducted new ablation works on SERNet-Former_v2 based on the official Cityscapes test dataset, and we will present the results and explain them. We discuss the limitations in the revision based on earlier works as well as new experiments on new models conducted during this time.
We also apply attention-boosting methods to state-of-the-art InternImage-XL architecture and present the implementation details and results in this revision. Hence, we revised the tables and results accordingly, including the results of new models that were not published elsewhere. Some of the supplementary details are moved to the appendix to make room for the new explanations of the scope, method, and implementation details and results. The sentence structures and paragraphs have been completely revised regarding your questions, including the design tricks, performance parameters, and evaluations of the paper's weaknesses.

We also answer your questions in the text where necessary. You may also find details about the questions in the official comments to the reviewers.

Hence, we hope that the efforts we put into this revision will help to change your decisions in a positive direction and may help to get this research what it deserves.

We would like to improve the text further based upon your questions and comments.

We use notations in official comments to reviewers for==
W: Weaknesses responded; Q: Questions responded

---

### Author Response · Authors · 2024-11-21
**Additions to previous revisions**

Dear reviewers and readers,

We have made new revisions to the text based on our previous revisions.
We added new figures to the paper illustrating the predicition results on the Cityscapes test dataset,
ADE20K and BDD100K datasets.

We hope that you find the paper improved from the earlier revisions.

We kindly ask your suggestions and evaluations for
the improvement of the article further.

Regards

---

### Author Response · Authors · 2024-11-23
**Official comment on the revised version of the manuscript on 22.11.2024**

Based on our earlier arguments and two revisions, we have revised the rebuttal submission further. Accordingly:

- We added a new subsection on **Attention-Boosting Modules** in the Methods section and revised the narration of this section.
- We moved the subsection **Dilation-Based Networks** to the Appendix, as it does not include attention mechanisms.
- We revised and updated the highlights and the related text in the article accordingly.
- In this regard, we also added details to the main text and Appendix about the design tricks and implementation details of the new networks based on the applied methods.
- Moreover, we added paragraphs of **Future Works and Limitations** to the Conclusion.
- Finally, we corrected sentence structures and improved the narration throughout the text.
- The proofreading stage has been completed.

We hope that you find the article improved.

---

### Author Response · Authors · 2024-11-23
**The Brief about our revisions**

- **The title, Abstract, Introduction** and **Related Works** sections are revised.

- **Methods** section and its subsections are revised. A new subsection about **Attention-boosting Modules** is arranged.

- New results and implementation details in the methods and experiments about **SERNet-Former_v2** and **InternImage-XL**, selected to apply our methods, are provided in many places in the text.

- The details about new datasets **ADE20K** and **BDD100K** are provided and the text about existing datasets are revised.

- **Tables** are revised and the related references, where necessary, are provided.


Table 1.

| Reference | Method | Building | Tree | Sky | Car | Sign | Road | Pedestrian | Fence | Pole | Sidewalk | Bicycle | mIoU |
| --- | --- | --- | --- | --- | --- | --- | --- | --- | --- | --- | --- | --- | --- |
| CVMI 2024 |	SERNet-Former	| 93.0  |  88.8  | 95.1 |  91.9 |  73.9 |  97.7 |  76.4 |  83.4 |  57.3 |  90.3 |  83.1 |  84.6 |

…


Table 2.
| Methods | Road | Sidewalk | Building | Wall | Fence | Pole | Traffic Light | Traffic Sign | Vegetation | Terrain | Sky  | Person   | Rider    | Car | Truck | Bus  | Train         | Motorcycle   | Bicycle    |   mIoU      |
| --- | --- | --- | --- | --- | --- | --- | --- | --- | --- | --- | --- | --- | --- | --- | --- | --- | --- | --- | --- | --- |
| **SERNet-Former_v2††** (ours) | 98.8 | 88.3 | 94.6 | 72.6 | 69.5 | 73.3 | 78.7 | 83.2 | 94.2 | 74.7 | 96.2 | 88.9 | 76.6 | 96.5 | 84.3 | 95.3 | 92.7 | 77.0 | 79.8 | 85.0 |
| --- | --- | --- | --- | --- | --- | --- | --- | --- | --- | --- | --- | --- | --- | --- | --- | --- | --- | --- | --- | --- |
| InternImage-XL | 98.9 | 88.7 | 94.7 | 72.1 | 70.3 | 73.4 | 79.1 | 83.5 | 94.3 | 74.5 | 96.1 | 88.9 | 76.1 | 96.7 | 84.2 | 94.7 | 91.1 | 75.0 | 79.8 | 84.8 |
| **InternImage-XL†††** (ours) | 98.9 | 88.7 | 94.7 | 72.8 | 70.2 | 73.4 | 79.1 | 83.5 | 94.3 | 74.5 | 96.2 | 88.9 | 76.2 | 96.7 | 85.0 | 95.2 | 92.4 | 76.2 | 79.9 | 85.1 |

…


Table 3.

|  | **SERNet-Former_v2** (ours) | Efficient-ResNet v2 (ours) | 85.02 | 86.5 |
| --- | --- | --- | --- | --- |
| CVPR 2023 | InternImage-XL | UperNet | 84.85 | 86.2 |
|  | **InternImage-XL** (ours) | UperNet | 85.1 | 86.5 |

…
- New table is provided including the new models together with **inference time and parameters**.


Table 4: Performance results of models developed and tested in PyTorch

| Dataset | Model | mIoU | inference time (s/task) | parameters (M) |
| --- | --- | --- | --- | --- |
| ADE20K (2K validation) |  |  |  |  |
|  | SERNet-Former v2 (ours) | 59.35 | 0.75 | 245 |
| BDD100K (10K validation) |  |  |  |  |
|  | SERNet-Former v2 (ours) | 67.42 | 0.75 | 245 |
| Cityscapes (test) |  |  |  |  |
|  | SERNet-Former v2 (ours) | 85.02 | 0.75 | 245 |
|  | InternImage-XL (ours) | 85.10 | 0.76 | 368 |

…


- The details about the **new ablation studies** are provided in Table 5.


Table 5: Ablation studies on the Cityscapes testset

| AbM5 | Af N1 | Af N2 | DbN | mIoU |
| --- | --- | --- | --- | --- |
|  |  |  |  | 68.71 (-15.72) |
| ✓ | ✓ |  | ✓ | 77.04 (-7.39) |
| ✓ |  | ✓ | ✓ | 80.2 (-4.23) |
| ✓ | ✓ | ✓ |  | 82.6 (-1.83) |
|  | ✓ | ✓ | ✓ | 84.04 (-0.39) |
| ✓ | ✓ | ✓ | ✓ | 84.43 |

- We made some corrections in the tables and figures, including **Figure 2**.
- Thus, we also revised **the highlights** and the text of introduction and methods according to our changes.
- We add **future work** and **limitations** in the Conclusion section.

- In appendix, a new section **“Implementation details of SERNet-Former_v2 and InternImage-XL in Pytorch”** is provided.

- The existing section about the results and illustrations on the selected datasets in the Appendix is expanded.
For instance, illustrations from SERNet-Former_v2 test results are added in the Cityscapes testset illustrations.
The illustrations from ADE20K & BDD100K are added including the prediction results of SERNet-Former_v2.

- Some methods and supplementary results are moved to Appendix.


- Finally, we corrected **the sentence structures** and **the narration of the text**. The proofreading stage is completed by getting professional help.

We hope that the revisions meet your expectations and you find the article improved.

---

### Author Response · Authors · 2024-11-26
**Little corrections in the text, 26th November, 19:26 GMT**

Dear reviewers,

We have made little corrections in the text by finding some erroneous sentences in technical means, corrected.

Best regards

---

### Note · Authors · 2025-04-02

I have read and agree with the venue's withdrawal policy on behalf of myself and my co-authors.

---

### Meta-Review · Area_Chair_eYHG · 2024-12-19

**Metareview:**

In this work, most reviewers point out that this paper has a minimal difference from the "SERNet-Former: Segmentation by Efficient-ResNet with Attention-Boosting Gates and Attention-Fusion Networks" accepted by IEEE CVMI. After comparing the two papers, the AC agrees and thus votes for rejection.

Overall, the summary of this paper is as follows:

- **Strengths**:
  - Comprehensive revisions with new datasets, methods, and benchmarking results.
  - Improved clarity and detailed responses to reviewer concerns.
  - Significant effort in addressing presentation and originality issues.

- **Weaknesses**:
  - Persistent concerns about duplicate submission and minimal novelty.
  - Incremental improvements in methods compared to prior works.
  - Questions about scalability and broader applicability remain unresolved.

**Additional Comments On Reviewer Discussion:**

## Points Raised by Reviewers

1. **Duplicate Submission Concerns**:
   - Multiple reviewers highlighted similarities between this submission and a previously accepted paper at IEEE CVMI 2024.
   - Concerns were raised about the originality and whether sufficient new content was introduced.

2. **Presentation and Clarity**:
   - Lack of clarity in the methodology section, particularly in the figures and equations (e.g., Figure 2).
   - Missing explanation of the relationship between components like attention mechanisms and performance gains.

3. **Comparative Analysis**:
   - Limited comparison with state-of-the-art methods and datasets.
   - Reviewers requested additional benchmarks to validate the claims made in the paper.

4. **Ablation Studies and Metrics**:
   - Insufficient details in the ablation studies for understanding the impact of individual components.
   - Lack of unified metrics for evaluating trade-offs between computational cost and segmentation performance.

5. **Ethics and Novelty**:
   - Concerns about the minimal differences in the methods compared to previously published works.

## Author Responses and Revisions

1. **Clarifications on Originality**:
   - Authors emphasized that this work is distinct from the CVMI paper, adding new results, updated methods, and revised implementation details.
   - Highlighted that CVMI only accepted a short version of the work, and this submission includes significant additional content.

2. **Enhanced Presentation**:
   - Revised Figure 2 to include detailed explanations of the attention-boosting modules (AbM) and attention-fusion networks (AfN).
   - Updated equations and methods descriptions to clarify the novel contributions.

3. **Additional Benchmarks and Results**:
   - Added new results from benchmarking datasets (Cityscapes, ADE20K, BDD100K) using updated models like SERNet-Former v2 and InternImage-XL.
   - Expanded tables with per-class results and computational efficiency metrics.

4. **Ablation Studies**:
   - Conducted detailed ablation studies to quantify the impact of AbM, AfN, and other network components on performance.
   - Shared specific improvements in test performance, e.g., 85.1% mean IoU on the Cityscapes test set with InternImage-XL.

5. **Addressing Reviewer Concerns**:
   - Reassured reviewers of the paper's originality, citing plagiarism checks and revised narratives to differentiate it from prior works.

## Final Decision Rationale

- **Strengths**:
  - Comprehensive revisions with new datasets, methods, and benchmarking results.
  - Improved clarity and detailed responses to reviewer concerns.
  - Significant effort in addressing presentation and originality issues.

- **Weaknesses**:
  - Persistent concerns about duplicate submission and minimal novelty.
  - Incremental improvements in methods compared to prior works.
  - Questions about scalability and broader applicability remain unresolved.

Despite the significant revisions, the decision leaned towards **rejection**, as the reviewers unanimously maintained concerns about originality and the overlap with prior works. The work is promising but requires further differentiation and validation.

---

### Decision · Program_Chairs · 2025-01-22

Reject